# IN SEARCH OF THE LONG-TAIL: SYSTEMATIC GENERATION OF LONG-TAIL KNOWLEDGE VIA LOGICAL RULE GUIDED SEARCH

## ABSTRACT

Since large language models (LLMs) have approached human-level performance on many tasks, it has become increasingly harder for researchers to find tasks that are still challenging to the models. Failure cases usually come from the long-tail distribution – data to which an oracle language model could assign a probability on the lower end of its distribution. Systematically finding evaluation data in the long-tail distribution is important, but current methodology such as prompt engineering or crowdsourcing are insufficient because coming up with long-tail examples is also hard for humans due to our cognitive bias. In this paper, we propose a **Logic-In**duced-**K**nowledge-Search (**LINK**⚲) framework for systematically generating long-tail knowledge statements. Grounded by a symbolic logic rule, we search for long-tail values for each variable of the rule by first prompting a large language model, then verifying the correctness of the values with a critic, and lastly pushing for the long-tail distribution with a reranker. Using this framework we construct a dataset, **Logic-In**duced-Long-**T**ail (**LINT**)[1], consisting of 200 symbolic rules and 40K knowledge statements spanning across four different domains. Human annotations find that 89% of the statements in LINT are factually correct. In contrast, ChatGPT and GPT4 struggle with directly generating long-tail statements under the guidance of logic rules, each only getting 61% and 79% of their statements correct. Moreover, their "long-tail" generations in fact fall into the higher likelihood range, and thus are not really long-tail. Our findings suggest that LINK is effective for generating data in the long-tail distribution while enforcing quality. To demonstrate how the community can utilize LINT for systematically evaluating LLMs' capabilities in the long-tail distribution, we challenge the models with a simple entailment classification task using samples from LINT. We find that ChatGPT and GPT4 performances drop by 2% and 4% when reasoning on long-tail knowledge statements compared to on head distribution statements. We hope our work can inspire future research on generating evaluation data in the long-tail distribution.

## 1 INTRODUCTION

Scalable Oversight (Bowman et al., 2022) – the challenge of supervising systems that potentially outperform us on most skills - seems more imminent than ever, when large language models (LLMs) have approached broadly human-level performance on many tasks (OpenAI, 2023; Ouyang et al., 2022). Systematically finding examples that are still challenging to the models has become increasingly hard, and existing works that do find such examples (Bubeck et al., 2023; Borji, 2023; Kocoń et al., 2023) mainly utilize prompt engineering and crowdsourcing. The failure cases of LLMs usually come from the long-tail distribution: a sample falls in the long-tail distribution if it is on the lower end of the distribution of an oracle language model. Since an oracle model is inaccessible, we can approximate the long-tail distribution using the existing largest model that outputs log probability, InstructGPT (`text-davinci-003`).

---

[1] https://doi.org/10.5281/zenodo.10126934

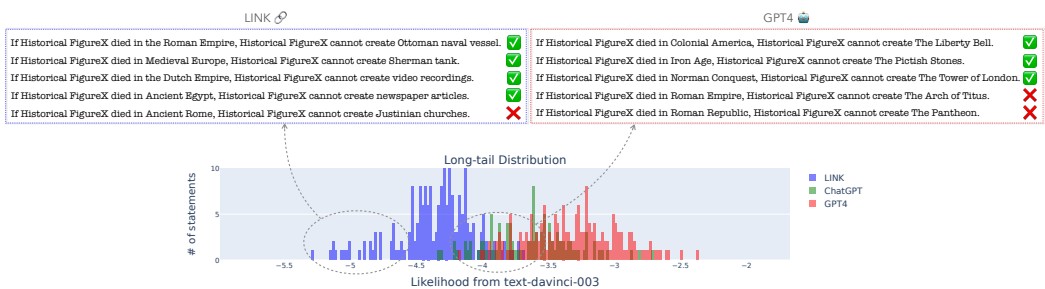

Figure 1: We demonstrate the distribution of intended long-tail generations from LINK(✐) and LLM baselines (ChatGPT and GPT4) for the same symbolic rule. LINK generations clearly lies in a more long-tail distribution than the baseline generations. In the bottom 20% (40 statements) of each distribution, LINK only made 9 factual errors, while GPT4 made 18 factual errors. We randomly sample 5 statements each from the bottom 20% of LINK(✐) and GPT4(🤖)'s long-tail distribution generation.

Current methodologies have become insufficient for systematically generating data in the long-tail distribution, because they are heavily dependent on human creativity, and thus also confined on human cognitive bias (Tversky & Kahneman, 1973; 1974), which makes it hard to come up with infrequent scenarios and make novel associations (Kray et al., 2006). Generating data in the long-tail distribution is fundamentally hard for LLMs as well. Since they have been trained to generate the "most likely" next token given the context, they may fall back to the head distribution, or even worse, hallucinate (Mündler et al., 2023).

Long-tail data is important for evaluating competent LLMs, yet so far there has not been any systematic exploration of generating long-tail evaluation data. In this paper, we would like to nail down on the generation of long-tail statements. We propose a generation framework, **Logic-In**duced-**K**nowledge-Search (**LINK**✐), where the statements are grounded in symbolic rule templates which consist of variables and predicates that regulate the relationship between variables. These templates serve as the foundation of the statements, as the statements are guaranteed to be correct if the values of the variables satisfy all predicates in the rule. We verbalize the symbolic rules into natural language statements by searching for appropriate values for the variables from LLMs. The entire process is named as **knowledge beam search** (Figure 2), as it is inspired from the sequential beam search. In knowledge beam search, we search for values of the variables one-by-one in a designated order. We construct an instruction prompt for each variable that consists of the variable and the predicates it is in, and then we prompt a large language model for values. For each variable, we ensure quality of the values using a critic model that checks for data type conformity and factual correctness, and ensure that the values fall into the long-tail distribution using a reranker model. Our logic-grounded generation framework allows us to safely wield the creative power of LLMs to systematically reach the long-tail distribution without suffering from hallucination.

Using LINK, we curate a dataset, **Logic-In**duced-Long-**T**ail (**LINT**), which contains 200 symbolic rules alongside with 20K+ long-tail knowledge statements spanning across four different domains. We also release 20K+ knowledge statements in the head distribution obtained using the same framework as an extended resource for studying LLMs' capability in different distributions. The knowledge statements are presented in a `premise,conclusion` format.

We find that it is indeed hard for LLMs to generate statements in the long-tail distribution by following instructions. We prompt ChatGPT and GPT4 to directly generate long-tail knowledge statements from the symbolic rules we curated in LINT. The models are presented with an instruction that includes a symbolic rule and are specifically asked to generate long-tail values for all variables in the rule. The generations of the models end up falling into the distribution with high log probability. i.e. the head distribution (Figure 1), which means that models are unable to directly retrieve knowledge from the long-tail distribution. In addition, human annotators find that compared to LINT, LLMs' generations more frequently contain factual errors or wrong data types.

LINT can be a useful resource for the community to systematically generate challenging datasets that test LLMs' capabilities in the long-tail distribution. We use a subset of LINT to create a simple task where LLMs are asked to reason on the knowledge statements by classifying entailment between the

premise and conclusion. ChatGPT and GPT4's capability in identifying incorrect knowledge drop by $\sim$3% in the long-tail distribution. In addition, we evaluate the models on diverse probing templates corresponding to each statement, and the models show significant bias on different template formats.

LINK is our first step towards the generation of long-tail data. The biggest challenge of generating long-tail data for LLMs is juggling enforcing factual correctness with remaining in lower data likelihood, and our logic-induced knowledge beam search framework provides a satisfying solution to this problem. Our simple experiment using LINT also proves that long-tail data found using our framework is indeed challenging for LLMs. We hope our work serves as a starting point of the series of work on systematically generating data in the long-tail distribution, and that our dataset can be a useful resource for evaluating LLMs in the long-tail distribution.

## 2 LOGIC-INDUCED-KNOWLEDGE-SEARCH (LINK⌲)

### 2.1 DEFINITION OF LONG-TAIL KNOWLEDGE STATEMENTS

Long-tail statements can exist in various length and forms. As the first step of exploring the long-tail distribution, in this work we scope knowledge statements under a `premise, conclusion` structure, which is a common way of knowledge representation (Sap et al., 2019). More specifically, the premise introduces some factual knowledge about a person or an object, and the conclusion is some information about the person or the object that can be deduced from the premise. Although it is hard to define the exact boundary between the long-tail distribution and the head distribution, we can still define them comparatively: for a group of statements with similar length and format, statements in the long-tail distribution should have lower likelihood than those in the head distribution. When we generate long-tail statements, we aim to generate those that lie on the part of the distribution with lower likelihood, compared to the rest of the statements of similar length and format.

Intuitively, for statements with similar length and format, the likelihood of content words in the statements heavily impacts the likelihood of the statement. Therefore, we can create long-tail statements from a symbolic rule and then fill in content words that are less likely to appear in the training corpus. Creating knowledge statements from symbolic rules have two benefits: (1) The symbolic rules are designed to be correct, so we alleviate the pressure of ensuring the deductive plausibility of the statement throughout the entire generation process. (2) We can break down the generation process into multiple steps, each of which is conditioned on only one variable in the rule, so we make the knowledge generation task much easier for the model. In addition, we can more easily maneuver the distribution of individual values than the entire sentence.

In the following section, we introduce our pipeline for generating long-tail knowledge statements from symbolic rules. We first describe our criteria for creating symbolic rules, then we illustrate our Knowledge Beam Search pipeline step by step (illustrated in Figure 2).

### 2.2 CREATING SYMBOLIC RULES

A symbolic rule consists of a premise and a conclusion. The conclusion is a single predicate, while the premise contains a set of predicates connected by & operators. Each predicate is a triple of a verb phrase, a subject and an object, and each variable in the symbolic rule has a designated data type. See example below:

| | | |
|---|---|---|
| *allergic_to(Person X, Allergen A)* | | |
| *& one_type_of(Ingredient Z, Allergen A)* | $\rightarrow$ | *cannot_eat(Person X, Dish B)* |
| *& ingredient_in(Ingredient Z, Dish B)* | | |

We have a few criteria for creating symbolic rules:

1. The symbolic rule should be topically referring to a person or an object, and the premise and conclusion of the symbolic rule need to have the same subject. This is to control the scope of the symbolic rule and decrease the verification difficulty.

2. The symbolic rule should be linearly chained. Starting from the subject in the premise, we should be able to find a chain of variables that connects the subject to the object in the

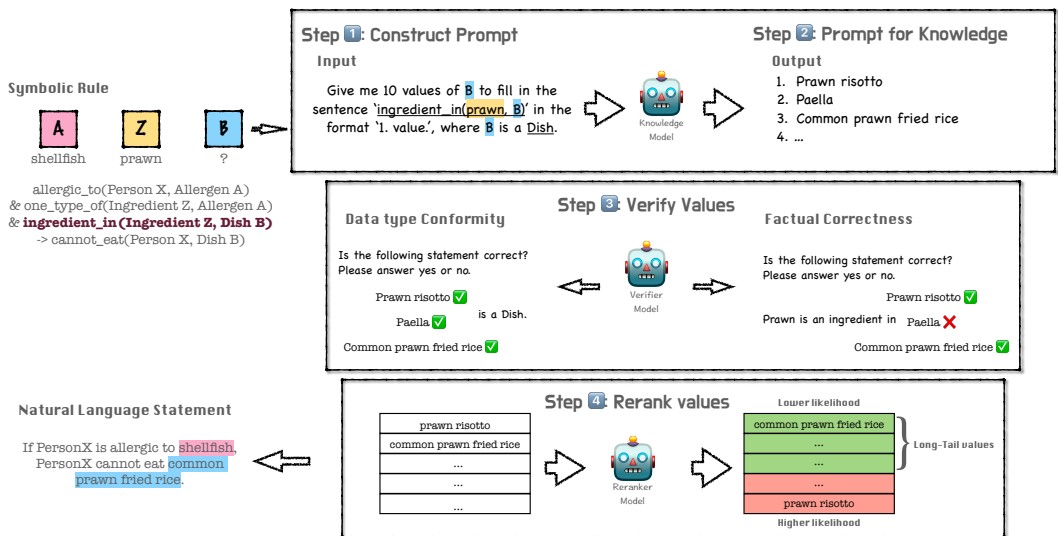

Figure 2: An overview of our knowledge beam search workflow(§ 2.3). Grounded on a symbolic rule, we search for values of the variables one-by-one in a designated order. This example demonstrates the search for Variable *B* conditioned on the values of *A* and *Z* from previous steps. We first construct a prompt as shown and prompt for values for *B*. Afterwards, we use a critic model to filter out the invalid values based on data type conformity and factual correctness. Lastly, we use a reranker model to control the distribution of the values. We populate the symbolic rule with the values and convert it into a natural language statement. Note that we only verbalize the predicates containing *Person X*, because all other predicates contain knowledge that the model should have.

conclusion (or vice versa), and there should not be any variable not included in this chain. This is to simplify the search process.

3. The premise and conclusion should not be paraphrases. For example, *allergic_to(Person X, Allergen A)→ reacts_badly_to(Person X, Allergen A)* is not a valid symbolic rule. This is to ensure that the symbolic rule contains some reasoning.

4. The symbolic rule should contain at least 3 variables and 2 predicates in the premise. This is to ensure some degree of complexity in the symbolic rule.

5. The symbolic rule should not contain predicates that are potentially out of scope of LLMs' knowledge. For example, *has_height(Tree X, Height Y)* is not a valid predicate, because it is unlikely that LLMs have knowledge about the exact height of a specific type of tree, which will likely lead to hallucination.

We create symbolic rules that spans across four domains: temporal, locational, food and physical conditions, and natural properties. In total, we create 65 person-related rules and 136 object-related rules. The detail of creating symbolic rules is in Appendix A.

## 2.3 KNOWLEDGE BEAM SEARCH

**Defining search order.** Since the symbolic rules are linearly chained, we can search all variables one by one without repetition. We always start with the subject of the sentence – the person or the object. We call this variable "the generic node", as the exact value is generically named *DatatypeX*. In the rule in § 2.2, for example, *Person X* is the generic node we start with. Starting from the generic node in the premise, we find a chain of variables that connects the subject to the object in the conclusion: *X, A, Z, B*.

For some rules that call for factual knowledge in numeric format, such as Age, Height, Year, etc., we empirically find that it increases the search quality to start from the generic node in the conclusion and connect to the object in the premise.

**Constructing Prompt.** For each variable, we construct the search prompt with all predicates that contain the variable and other previously searched variables. For example, for the variable *B* in the rule in § 2.2, the predicate in the premise that contains *B* is *ingredient_in(Ingredient Z, Dish B)*, and we haven't searched the conclusion yet. Since *Z* is searched before *B*, we assume *Z* takes the value of "prawn", so we construct the prompt as follows:

*Give me 50 values of B to fill in the sentence "ingredient_in(prawn, B)" in the format "1. value.", where B is a Dish.*

**Prompting InstructGPT for knowledge.** Given a partially searched beam that consists of values of the searched variables and the current variable, we want to obtain 200 values of the current variable from the knowledge model. Given the prompt that restricts the model to only generate 50 values, we call the API 4 times. We set the temperature to 0.7 to encourage diversity. After each call, we verify the values using a critic model (see paragraph below). To prevent duplicates in subsequent calls, we explicitly request in the prompt not to generate verified correct values, and we modify the token probability of incorrect values to ban regenerating them. We implement an early stop mechanism: if for two consecutive calls we are unable to get any correct values, we terminate the search for the beam.

**Verifying values using a critic model.** We use `Flan-T5-XXL` as the critic that checks data type conformity and factual correctness of the searched values because the model has been instruction-finetuned and can be used zero-shot. We ask the model to output *yes/no* on the correctness of a given statement. For data type conformity, the statement is "{*value*} *is a* {*data type*}." For factual correctness, we convert the symbolic predicate into a natural language statement. We obtain the *yes* token probability and implement a dynamic critic threshold that adjusts the threshold for accepting values for different predicates. For details of the implementation and evaluation of the critic model, see Appendix B.

**Pushing values to head or long-tail distribution using a reranker model.** After searching for each variable, we use `llama-7B` as a reranker on the (partially or fully) searched beam, as an estimate of the distribution the particular combination of variable values falls in. We convert a beam into a natural language sentence by transforming each predicate into a natural language statement (*ingredient_in(prawn, prawn risotto)* → *Prawn is an ingredient in prawn risotto*) and concatenating them with "and." We then obtain the sentence likelihood from `llama-7B`, and rank the sentences from *lowest* likelihood to the *highest* likelihood. We take top the 75% values unless there are more than 200 values, in which case we take the top 200 values. Then we move on to the next variable. Simultaneously, we also search for values in the head distribution. All processes remain the same except we rank the sentences from the *highest* likelihood to the *lowest* likelihood.

From 65 person-related rules and 136 object-related rules across four domains, we use Knowledge Beam Search to curate **L**ogic-**In**duced-Long-**T**ail(**LINT**) consisting of 27K+ long-tail knowledge statements. In addition, we also release 26K+ head distribution statements that are also searched with the LINK framework, as it could also be a valuable resource for comparing model behavior in the head and long-tail distribution.

## 3 TESTING MODEL KNOWLEDGE IN THE LONG-TAIL DISTRIBUTION

### 3.1 GENERATING LONG-TAIL STATEMENTS FROM SYMBOLIC RULES

In this section, we show that LLMs cannot satisfactorily generate long-tail statements directly from symbolic rules.

For each symbolic rule in LINT, we prompt ChatGPT and GPT4 to populate the rule with all variables simultaneously. The prompt includes a natural language version of the symbolic rule with the data types of the variables specified. Table 1 shows a symbolic rule and its corresponding prompt.

To ensure the long-tail distribution, we explicitly add *"Use less frequent terms of A and B and C"* in the prompt.

Table 1: An example symbolic rule and its corresponding prompt.

| | |
|---|---|
| Symbolic Rule | died_in(Historical Figure X, Historical Time Period A )
& was_created_during(Artifact B, Historical Time Period C)
& is_earlier_than(Historical Time Period A, Historical Time Period C)
→ cannot_create(Historical Figure X, Artifact B) |
| Prompt | In the following sentence, A is a Historical Time Period, B is a Artifact,
C is a Historical Time Period. Find values of A, B, C to fill in the blank
in the sentence 'If Historical Figure X died in [A] and [B] was created
during [C] and [A] is earlier than [C], then Historical Figure X cannot
create [B].' and make it a grammatical and correct sentence.
Give me 50 values in the format '1. A=, B=, C='. |

For each symbolic rule in LINT, we obtain 200 statements in the head distribution and 200 statements in the long-tail distribution from ChatGPT and GPT4 respectively. In the remainder of this section, we compare the quality of the model generated statements with the statements in LINT.

## 3.2 LLMs CANNOT REACH THE LONG-TAIL DISTRIBUTION WITH ONLY INSTRUCTIONS

We calculate the log likelihood of all statements generated by LINK, Chat-GPT and GPT4 from the symbolic rules in LINT, using InstructGPT as an approximation to the distribution of an oracle model, We calculate $\delta = mean(D(H)) - mean(D(L))$ for each set of statements generated from each symbolic rule, where $D(\cdot)$ means the log likelihood distribution of InstructGPT, $H$ is the set of statements intended to be the head distribution, and $L$ is the set of the statements intended to be in the long-tail distribution.

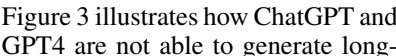

Figure 3: $\delta$ of the generation methods for all person-related rules and object-related rules.

Figure 3 illustrates how ChatGPT and GPT4 are not able to generate long-tail statements merely from prompting. Each grid on the x-axis represents a unique symbolic rule, and each grid on the y-axis represents $\delta \in [-1.5, 2]$. A $\delta$ close to 0 means that the intended head and long-tail distribution are inseparable, being larger than 0.3 empirically means a decent separation, and being negative means the intended head and long-tail distributions are reverted on the log likelihood scale of InstructGPT.

Averaged across 200 rules, LINT has a positive $\delta$ of 0.48, while Chat-GPT and GPT4 each has a delta of -0.14 and -0.02. The $\delta$ values for LINT (blue line) float above 0 for most of the rules, some even being above 0.5. On the other hand, the $\delta$ values for ChatGPT (red line) and GPT4 (green line) mostly locate around 0, with many rules having negative $\delta$ values. These low and negative $\delta$ values indicate that LLMs' generations are in fact not in the long-tail distribution, and sometimes even in the head distribution.

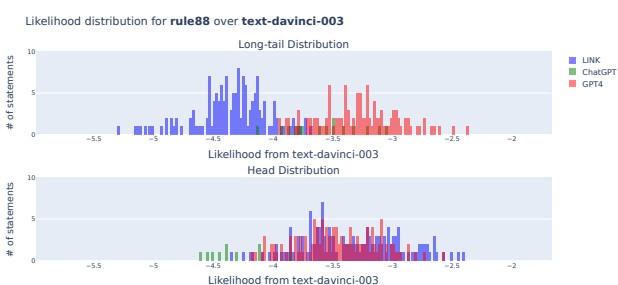

Figure 4: Only LINK generations fall in the correct distributions on the log likelihood scale of InstructGPT.

To better illustrate the distribution of statements generated by ChatGPT and GPT4, we sample a symbolic rule and plot the log likelihood of InstructGPT on the generated statements from the three

methods Figure 4. To rule out the effect of incorrect statements on the distribution, we only plot the statements that are marked as correct in human evaluation (explained in § 3.3). The likelihood distribution of more rules can be found in Appendix C.1.

The long-tail statements from LINK clearly fall in a much lower probability distribution than GPT4's generations. Moreover, GPT4's generation in the "long-tail distribution" in fact falls in the same distribution as its head distribution, indicating that simply using instructions in prompts are not sufficient to guide the model to generate long-tail statements. This finding highlights the importance of using a reranker on each variable in LINK.

### 3.3 LLMs have Lower Data Type Conformity and Factual Correctness

In addition to distribution correctness, we also evaluate the data type conformity and factual correctness of LLMs' long-tail knowledge generations using crowdworkers from Amazon Mechanic Turk[2] (AMT). For data type conformity, we ask an AMT worker *Is {variable} a {data type}?* for each variable in the symbolc rule. For factual correctness, we ask an AMT worker *Does the premise entail the conclusion?*. We uniformly sample 4,000 statements from all statements in LINT for human evaluation, of

Table 2: LINK has both the highest factual and data type accuracy in human evaluation.

| Accuracy | ChatGPT | GPT4 | LINK |
|---|---|---|---|
| Data Type | 85.40 | 91.80 | **94.23** |
| Factuality | 67.50 | 84.82 | **88.71** |
| Overall | 56.44 | 78.23 | **83.95** |

which 2,025 are from head distribution and 1,975 are from long-tail distribution. We take 3 annotations for each statement and use a majority voting strategy. The agreement of annotations can be found in Appendix G.3. The AMT template can be found in Appendix G.2.

Table 2 shows that both ChatGPT and GPT4 underperform LINK in both data type conformity and factual correctness. Both models struggle more with enforcing factual correctness, a foreseeable behavior in the low likelihood realm. This finding highlights the effectiveness of the zero-shot critic model in LINK on improving factual correctness.

### 3.4 Discussion

Our analysis above has shown that by simply prompting ChatGPT and GPT4, arguable the state-of-the-art LLMs on language tasks, cannot effectively generate knowledge statements that are both in the long-tail distribution and correct. Our approach LINK, albeit using a weaker LLM as the knowledge model, is able to achieve the long-tail distribution and correctness simultaneously with the help of smaller open source models. While utilizing our LINK, one may easily substitute InstructGPT with stronger LLMs as knowledge models and get even better search results.

## 4 Using LINT for Model Evaluation

We hope LINT can be a useful resource for the community wishing for systematically evaluating LLMs' capabilities in the long-tail distribution. In this section, we provide our evaluation on a simple **entailment classification task** to motivate further usage of our dataset.

We used all human evaluated knowledge statements that are labeled correct or factually incorrect in LINT, with 1,925 statements in head distribution and 1,856 statements in long-tail distribution. Each statement is converted into 13 probing question to avoid template bias affecting the probing results, where each probing template corresponds to either a positive label ("Yes", "True", or "Right") or a negative label ("No", "False", or "Wrong"). The probing templates are summarized in Appendix E.1.

We experiment on three LLMs: InstructGPT, ChatGPT and GPT4. We use data labeled as both factually correct and having correct data type as positive samples and use data labeled as factually incorrect during human evaluation as negative samples. For all positive samples, entailment would form between the premise and conclusion, while for negative samples, non-entailment would form. Given that LINT has a 84% accuracy, we have more positive samples than negative samples. Out

---

[2]https://www.mturk.com

Table 3: Performance on the entailment classification task of three LLMs and human baseline decrease on the long-tail distribution compared to the head distribution. LLMs also have a template bias towards negative templates – when the target token is "no", "false" and "wrong".

| Domain | Distribution | InstructGPT | | ChatGPT | | GPT4 | | Human Baseline |
|---|---|---|---|---|---|---|---|---|
| | | Zero-Shot | COT | Zero-Shot | COT | Zero-Shot | COT | |
| Temporal | **Head** | | | | | | | |
| | - Positive Template | 12.12 | 47.96 | 45.28 | 59.83 | 79.43 | 87.03 | - |
| | - Negative Template | 88.23 | 79.33 | 83.57 | 80.35 | 86.20 | 79.81 | - |
| | All | 53.10 | 64.85 | 65.90 | 70.88 | 83.07 | 83.14 | 84.69 |
| | **Longtail** | | | | | | | |
| | - Positive Template | 16.56 | 47.01 | 44.29 | 58.72 | 77.41 | 81.99 | - |
| | - Negative Template | 83.89 | 76.02 | 79.90 | 76.92 | 83.08 | 76.39 | - |
| | All | 52.81 | 62.63 | 63.46 | 68.52 | 80.46 | 78.97 | 83.20 |
| Food and Physical Conditions | **Head** | | | | | | | |
| | - Positive Template | 22.61 | 52.85 | 61.84 | 70.38 | 74.64 | 78.46 | - |
| | - Negative Template | 84.80 | 71.65 | 77.66 | 74.91 | 86.93 | 84.63 | - |
| | All | 56.10 | 62.97 | 70.36 | 72.82 | 81.26 | 81.78 | 83.83 |
| | **Longtail** | | | | | | | |
| | - Positive Template | 27.66 | 55.38 | 67.01 | 68.77 | 72.87 | 80.89 | - |
| | - Negative Template | 78.22 | 69.71 | 74.19 | 74.24 | 82.87 | 79.60 | - |
| | All | 54.88 | 63.09 | 70.88 | 71.71 | 78.25 | 80.19 | 85.13 |
| Natural Properties | **Head** | | | | | | | |
| | - Positive Template | 3.47 | 18.52 | 33.54 | 35.95 | 31.33 | 50.13 | - |
| | - Negative Template | 93.85 | 70.75 | 72.68 | 60.96 | 89.05 | 81.58 | - |
| | All | 52.13 | 46.64 | 54.62 | 49.41 | 62.41 | 67.07 | 82.31 |
| | **Longtail** | | | | | | | |
| | - Positive Template | 3.71 | 16.97 | 33.39 | 34.09 | 24.62 | 44.53 | - |
| | - Negative Template | 93.84 | 70.50 | 71.21 | 60.13 | 87.36 | 79.57 | - |
| | All | 52.24 | 45.79 | 53.75 | 48.11 | 58.40 | 63.40 | 82.45 |
| Locational | **Head** | | | | | | | |
| | - Positive Template | 22.86 | 52.86 | 51.67 | 63.10 | 52.14 | 59.29 | - |
| | - Negative Template | 87.55 | 65.92 | 68.98 | 61.84 | 75.71 | 77.14 | - |
| | All | 57.69 | 59.89 | 60.99 | 62.42 | 64.84 | 68.90 | 75.71 |
| | **Longtail** | | | | | | | |
| | - Positive Template | 41.39 | 57.30 | 51.87 | 57.68 | 59.74 | 63.30 | - |
| | - Negative Template | 76.24 | 62.12 | 63.40 | 58.11 | 73.03 | 70.30 | - |
| | All | 60.16 | 59.90 | 58.08 | 57.91 | 66.90 | 67.07 | 67.42 |

of the 13 question templates, some aim for positive target tokens and others aim for negative target tokens, so random guessing performance is 54% instead of 50%.

In addition to zero-shot prompting, we also use a variation with Chain-of-Thought (COT) (Wei et al., 2022) prompting. For each question, we require the model to generate a rationale with "*Let's think step by step*". After that, we ask for the final answer. We also provide two COT examples to form the demonstration context for in-context learning (ICL), randomly shuffling the order of an example with a positive template and another one with a negative template.

We obtain a human performance baseline on the same set of statements. We recruit 17 AMT workers who do not participate in the evaluation task (and thus have not seen the task data). The workers see the knowledge statements in `premise, conclusion` format and are asked to select "*yes/no*" to whether the premise entails the conclusion. The workers are asked to use search engines to verify their answers. See the AMT template in Appendix G.2.

For each domain, we report aggregated and template-wise performance of InstructGPT, ChatGPT and GPT4 as well as human baseline performance in Table 3.

**The negative template performance experience a "long-tail drop" for all models and domains.** InstructGPT, ChatGPT and GPT4 performance on negative templates on the long-tail distribution decrease by 3.32%, 2.79%, 2.8% respectively without COT, and 1.82%, 1.71%, 3.52% respectively with COT. This indicates that models perform worse at identifying incorrect knowledge statements in the long-tail distribution.

**The positive template performance experience a "long-tail rise" for all models and most domains.** For all models, the performance on positive templates is significantly lower than on negative templates. Besides "Natural properties" domain, we witness an increase in performance in the long-tail distribution on the positive templates, contrary to our observation on the negative templates.

We examine the rationale the models generate during COT in Table 8 and find that the models tend to avoid drawing a "definite conclusion". We hypothesize that this phenomenon may be a result of model adjustments made with Reinforcement Learning with Human Feedback (RLHF) (Ouyang et al., 2022), and these model adjustments may have a weaker effect on the long-tail distribution due to the knowledge being less frequent during training. This hypothesis can be further supported by the significant increase in positive template performance after COT prompting, where COT weakens the model's original distribution over the answer tokens in the in-context examples. We leave the exact explanation to future work.

**Human Performance do not drop for long-tail distribution.** Performance drop in the long-tail distribution does not happen in human baseline for 3 out of 4 domains, with the locational domain as the exception. This is expected because humans can verify their knowledge using search engines, so infrequent knowledge does not challenge humans as much as models.

## 5 RELATED WORK

Since the introduction of competent LLMs such as ChatGPT and GPT4, researchers have been **finding failure cases** using prompt engineering and crowdsourcing (Bubeck et al., 2023; Borji, 2023; Kocoń et al., 2023; Mao et al., 2023), or focused on analyzing large language model performance on a specific domain such as solving math problems (Davis & Aaronson, 2023), answering medical questions (Nori et al., 2023), or generating code (Poldrack et al., 2023).

The community has realized the importance of testing language models' abilities in the long-tail distribution (Godbole & Jia, 2022), and more works focus on **generating less common data for probing LLMs**. RICA (Zhou et al., 2020) proposes to include novel entities in self-contained commonsense statements to evaluate robust inference capabilities. UnCommonSense (Arnaout et al., 2022) proposes to evaluate models on informative negative knowledge about everyday concepts in addition to positively expressed commonsense knowledge. Razeghi et al. (2022) observe a correlation between the model performance on math problems and the frequency of numeric and temporal terms from those instances in the pretraining data.

In addition to probing models' reasoning ability on less common data, recent works also propose to **test language models' ability to generate less common data**. Chen et al. (2023) propose a negative knowledge generation task where models generate uncommon knowledge with negation conditioned on constrained keywords. Tang et al. (2023) introduce the "less likely brainstorming" task that asks a model to generate outputs that humans think are relevant but less likely to happen.

Generating uncommon data is not only challenging for language models, but also **challenging for humans because of our cognitive bias**. Tversky & Kahneman (1974) observe that humans are prone to more systematic errors when facing uncertain events, and Tversky & Kahneman (1973) reveal that humans tend to evaluate the frequency of classes or the probability of events by availability, i.e., by the ease with which relevant instances come to mind. These traits make it difficult for humans to come up with novel associations (Kray et al., 2006), a crucial ability to create data in the long-tail distribution.

## 6 CONCLUSION AND FUTURE WORK

This paper is situated in the problem of generating challenging evaluation data for LLMs in the long-tail distribution. We proposed a long-tail data generation framework, LINK, which grounds the long-tail statements in symbolic rule templates. Using LINK, we created a dataset, LINT, that contains 200 symbolic rules and 20K+ long-tail knowledge statements. We showed that directly generating long-tail knowledge statements is still hard for the state-of-the-art LLMs, and our approach LINK is effective for reaching the long-tail distribution while preserving factual correctness. In the future, we plan to extend LINK to support generating from more diverse and complex symbolic rules.

Our dataset LINT is a useful resource for curating challenging evaluation data for LLMs. We exemplified the effectiveness of LINT using a simple entailment classification task, and showed that our long-tail distribution data is more challenging to LLMs than the head distribution data. Future work may explore more challenging tasks such as knowledge base completion and multi-hop reasoning.

## REPRODUCIBILITY STATEMENT

**Algorithm.** We provide accurate description of our LINK framework in Section 2, with additional details on creating symbolic rules in Appendix A and on critic model in Appendix B. We include details of the entailment classification task in Appendix E.1.

**Prompt Engineering.** The prompts we used for conducting Knowledge Beam Search, generating long-tail statements with LLMs and creating symbolic rules are described and exemplified in the main paper.

**Data and Source Code.** The symbolic rules we use for LINK, all data generated by LINK, the human annotation data on LINT and the entailment classification task data are released on Zenodo [https://doi.org/10.5281/zenodo.10126934]. Data generated by ChatGPT and GPT4 as well as their human annotation will be available to the public upon request. Source code for creating symbolic rules, building LINK and conducting entailment classification task will be released publicly after the anonymity period.

**Crowdsourcing.** Templates for generation quality evaluation task and entailment classification task are included in Appendix G.2. Annotator agreement statistics is included in Appendix G.3.

## ETHICS STATEMENT

**Data.** All data we collected through LLMs in our work are released publicly for usage and have been duly scrutinized by the authors. Data for all human studies that we conduct are also publicly released with this work, with appropriate annotator anonymizations.

**Crowdsourcing.** All our crowdworkers are from countries where English is the primary language and recruited from Amazon Mechanic Turk. For all our human studies, the task is set up in a manner that ensure that the annotators receive compensation that is above minimum wage ($15/hour). Since we conduct extensive qualification tasks before annotations, crowdworkers that participate in the qualification are compensated more than minimum wage, given the time taken to read and understand task instructions and examples. Furthermore, we ensure that we correspond with crowdworkers over email to address their queries. Crowdworkers have also been given bonuses for providing feedback to the task, or consistently providing good-quality annotations.

**Potential Use.** Our framework LINK may only be used for generations that follow the ethics guideline of the community. Using LINK on mal-intention-ed rules or searching for toxic and harmful values is a potential threat, but the authors strongly condemn doing so.

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

## A  Symbolic Rule Creation

Following the criteria mentioned in § 2.2, we come up with 14 meta-rules. These meta-rules are centered on persons or objects and cover four main domains. To diversify our symbolic rules, we prompt GPT4 to specify the data types of variables and embellish the predicates based on the specified data types. Specifically, for the variable we want to specify, we first prompt GPT4 to get 10 potential data types. We manually filter out those invalid data types, such as some overly fine-grained ones that can be considered as values instead of data types. Then for each specified data type, we will prompt the GPT4 to find the best verb that can replace the original general one and translate it into natural language.

Table 4 shows an example of one meta-rule, related prompts for expansion and one expanded rule.

Table 4: An example meta-rule and its corresponding prompts for expansion.

| | |
|---|---|
| Meta-Rule | is_of_age(Person A, Age X) & requires_a_minimal_age_of(Object B, Age Y) & is_smaller_than(Age X, Age Y) → cannot_operate(Person A, Object B) |
| Prompt for Data Type Expansion | In rule "requires_a_minimal_age_of(Object B, Age Y) & cannot_operate(Person A, Object B)", B is a variable representing an object. List 10 subcategories of object that B could be that also make the rule true. |
| Expanded Data Types | Vehicle, Machinery, Alcohol, Firearm, Tattoo Equipment, Tobacco Product |
| Prompt for Verb Optimization | cannot_operate(Person A, Object B) is equal to [mask](Person A, Vehicle B). Write the best predicate that could fit in [mask] token. |
| Expanded Rule | is_of_age(Person A, Age X) & requires_a_minimal_operating_age_of(Object B, Age Y) & is_smaller_than(Age X, Age Y) → cannot_drive(Person A, Vehicle B) |

## B  Critic Model

We find that while the critic model usually verifies data type conformity with high accuracy, it often creates false negatives when verifying factual correctness. Moreover, even within false negatives that result from the same predicate, the correct values get higher `yes` token probabilities than the incorrect values. We hypothesize that while the critic model is less confident about certain knowledge because it is trained on a smaller portion of the knowledge than `text-davinvi-003`, it can still rank the values inherently. Therefore, we extract the probability of the `yes` token instead of taking the argmax. We also implement a dynamic critic threshold that adjusts the threshold for accepting values for different predicates. The algorithm is as follows:

1. We start with a threshold of 0.85.

2. If no correct values are found, we decrease the threshold by 0.05.

3. If some correct values are found, we set the threshold for the predicate to the current threshold and do not decrease it in further calls.

4. If the threshold is set but we find some values with a higher `yes` token probability than the threshold, we increase the threshold by an increment of 0.05 to accommodate the higher probability. Then we retrospectively reject previous accepted values with a lower `yes` token probability than the new threshold.

5. For data type conformity, we set a minimum threshold of 0.65 because we expect the model to be more confident.

In this way, we can find the maximum available threshold for each beam, which guarantees precision while reducing false negatives.

To verify the effectiveness of our critic model, we use crowd workers from AMT to evaluate the data type conformity and factual correctness of predicates. Specifically, for each symbolic predicate that contains two variables (e.g., *exist_during(Location X, Historical Time Period Y)*), we will present a

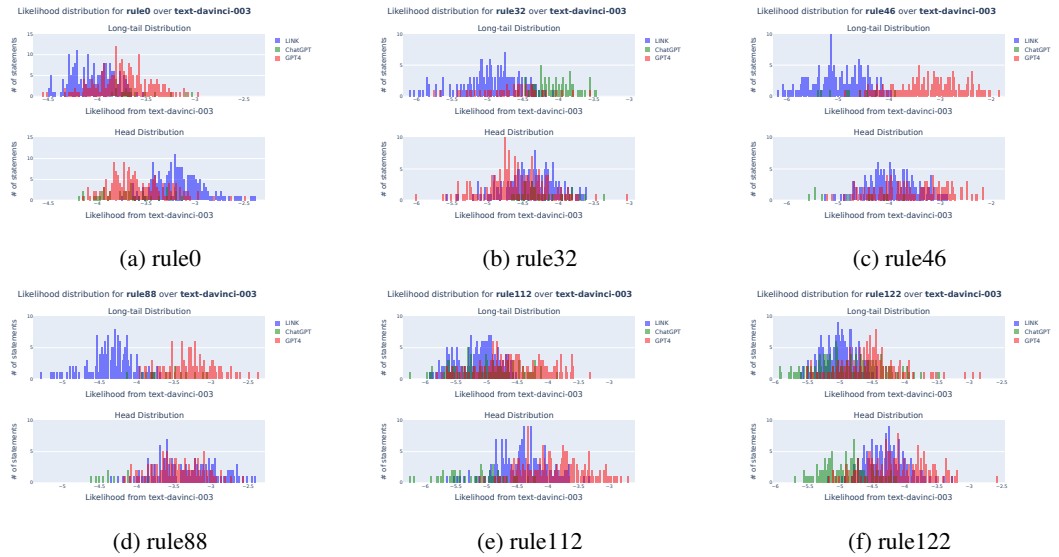

(a) rule0          (b) rule32          (c) rule46

(d) rule88          (e) rule112          (f) rule122

Figure 5: An illustration on the distribution of generated statements by LINK, ChatGPT and GPT4. While LINK's long-tail generations fall into a lower probability distribution than those of GPT4, GPT4's "long-tail distribution" overlaps with the head distribution, indicating that these generations are not truly long-tail.

statement in natural language (e.g., *Saigon* existed during *The Cold War*.) with 3 types of questions: (1) clear reference: Q1 and Q2. (2) factual correctness: Q3. (3) data type conformity: Q4 and Q5.

- **Q1:** Does "*Value A*" in the Statement "*Statement*" have a clear reference?
- **Q2:** Does "*Value B*" in the Statement "*Statement*" have a clear reference?
- **Q3:** Is the Statement "*Statement*" factually correct, with very high probability?
- **Q4:** Is the Statement "*Value A* is a *Data Type A*." factually correct, with very high probability?
- **Q5:** Is the Statement "*Value B* is a *Data Type B*." factually correct, with very high probability?

We sample 3 rules from our data and requested human annotators to rate the data type conformity and factual correctness of statements. Table 5 shows the error rate of each question. Only if all the questions are answered with "Yes" do we consider the statement as correct. The overall correctness of statements in head distribution and long-tail distribution are 0.8567 and 0.8467 respectively, which indicates a high quality of statements accepted by our critic model.

Table 5: Human verified data type conformity and factual correctness. Most errors occur on factual correctness.

|  | Q1 | Q2 | Q3 | Q4 | Q5 |
|---|---|---|---|---|---|
| Error Rate | 0.0004 | 0 | 0.0639 | 0.0011 | 0 |

## C ADDENDUM ON DISTRIBUTION

### C.1 ADDITIONAL DISTRIBUTION PLOTS FOR SYMBOLIC RULES

As an extension on § 3.2, we show the distribution of statements sampled by LINK, ChatGPT and GPT4 from 6 symbolic rules on InstructGPT (Figure 5).

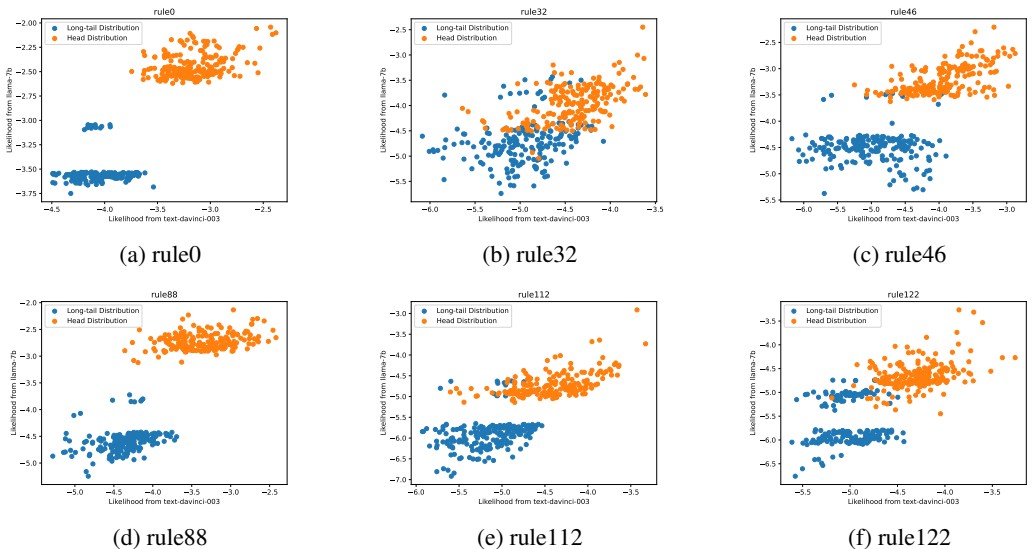

Figure 6: An illustration of the distribution comparison between `llama-7b` and `text-davinci-003` of generated statements by LINK.

## C.2 ACHIEVING THE LONG-TAIL DISTRIBUTION OF INSTRUCTGPT USING LLAMA-7B IN LINK

Given the size difference between Llama-7B and InstructGPT, we verify that using a smaller model to rerank statements is effective to select statements that fall in the long-tail distribution of a larger model. Figure 6 show that statements falling in the head distribution of Llama-7B also tend to fall in the head distribution of InstructGPT, and statements falling in the long-tail distribution of Llama-7B also tend to fall in the long-tail distribution of InstructGPT.

## D ABLATION STUDIES ON LINK

### D.1 EFFECT OF RERANKER ON LINK

To investigate the effectiveness of the reranker, we provide an ablation study on a few sampled rules by replacing the reranker step with a random sampling method. Figure 7 presents the distribution comparison of generated statements by LINK and the variant without the reranker. Without the reranker, the generated statements for both head distribution and long-tail distribution are pulled towards the center of the distribution, making them completely inseperable.

### D.2 INEFFECTIVENESS OF POST-HOC RERANKING FOR LLM GENERATED KNOWLEDGE.

To further highlight the importance of performing step-wise reranking in LINK, we confirm that applying a post-hoc reranker on the GPT4 generations from instructions does not have the same effect as LINK. We use the same reranking method as in LINK § 2.3 to rerank the generations from the *lowest* to the *highest* likelihood and take the top 75% results. For the head distribution, we rerank the generations from the *highest* to the *lowest* likelihood and take the top 75% results.

We evaluate on the same set of rules as in § 3.2 as an example. Figure 8 is an illustration of the distribution of generated statements by LINK, instruction-based GPT4 and instruction-based GPT4 with reranker. We observe that using post-hoc reranker still cannot achieve a separation between the generation of the head distribution and the long-tail distribution. It demonstrates that maneuvering the distribution during the searching process is necessary and more effective than post-hoc filtering.

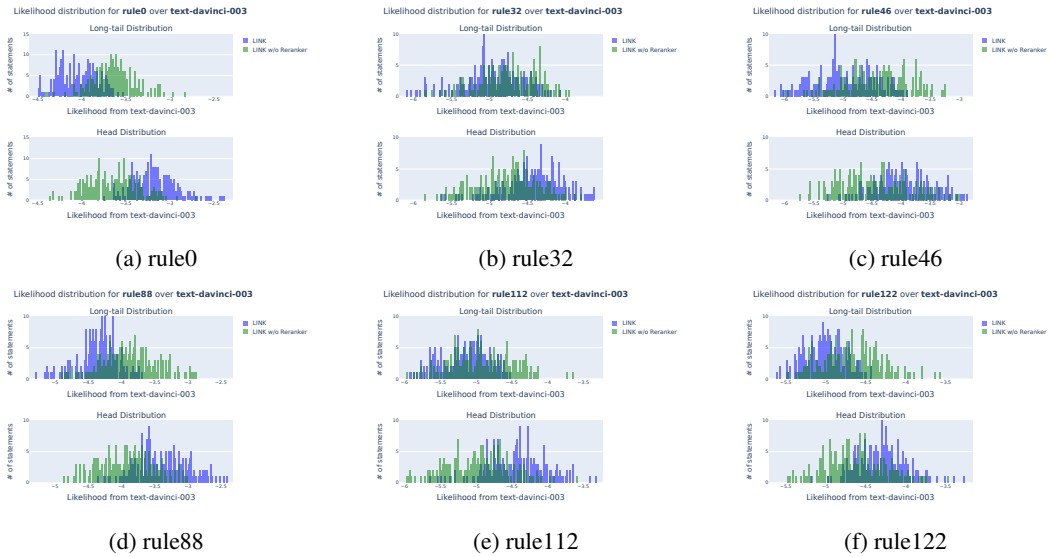

Figure 7: When we remove reranker from LINK, the distribution of the resulting head and long-tail statements are pulled towards the center. Using reranker is essential for separating the head and long-tail distribution.

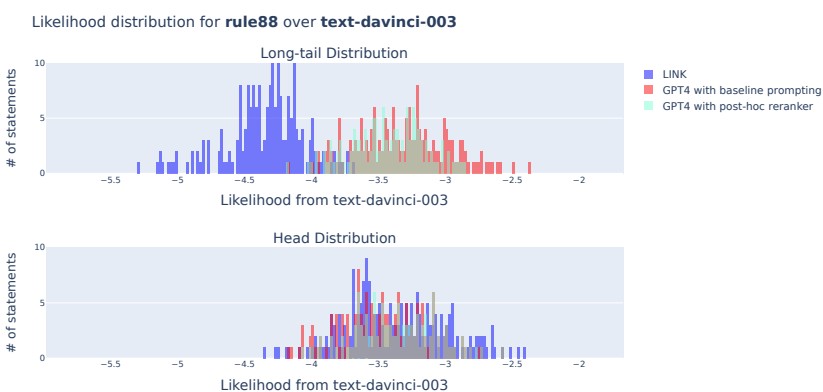

Figure 8: Post-hoc reranking of GPT4 does not help move the distribution towards the long-tail distribution.

### D.3 APPLYING GPT4 AS THE KNOWLEDGE MODEL

Using `text-davinci-003` as the knowledge model in LINK is due to our consideration of search cost, and one may replace it with stronger models and get better search results. As a proof of concept, we search on one rule (rule 88) using LINK with GPT4 and evaluate the data type conformity and factual correctness using crowdworkers.

Table 6 shows the generation quality of GPT4 using baseline prompting method, LINK and LINK with GPT4 as the knowledge model. Using a stronger model as the knowledge model can further improve the quality of generations by LINK. Figure 9 shows that whatever the knowledge model is, the distribution of generations by LINK can correctly fall in the long-tail distribution. Even though baseline GPT4 outperforms LINK on data type accuracy, factuality has a larger impact on the overall quality of the statements, to which baseline GPT4 underperforms LINK.

Table 6: The generation quality of LINK is further improved if using GPT4 as the knowledge model.

|  | Data Type | Factuality | Overall |
|---|---|---|---|
| GPT4 with baseline prompting | **96.00** | 68.82 | 66.47 |
| LINK | 78.00 | 94.00 | 74.00 |
| LINK with GPT4 | 81.00 | **98.00** | **79.00** |

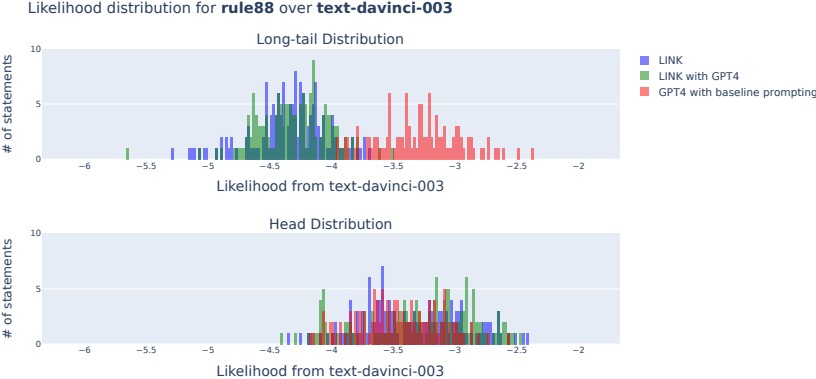

Figure 9: LINK using GPT4 creates statements that fall in a roughly similar long-tail distribution as the original LINK with InstructGPT.

## E ENTAILMENT CLASSIFICATION PROBING

### E.1 PROBING TEMPLATE

Table 7 shows templates we used for the entailment classification task. As mentioned in § 4, we divide the templates into positive templates and negative templates. Positive templates are those with a positive label (i.e., *Yes*, *Right* and *True*) and negative templates are those with a negative label (i.e., *No*, *Wrong* and *False*).

Most of the templates have definite labels across all rules. However, the label of Template 7 depends on the rules. If the rule has a positive conclusion (e.g., *Person X can use ChatGPT*), the answer to the question should be positive, i.e., *Yes*. On the contrary, if the rule has a negative conclusion(e.g., *Person X cannot use ChatGPT*), the answer to the question should be negative, i.e., *No*.

### E.2 RATIONALE ANALYSIS

As mentioned in § 4, we examine the rationale the model generated during COT in the entailment classification task and found that the models tend to avoid drawing a "definite conclusion". Table 8 shows an example.

## F DOMAIN-WISE STATISTICS OF LINT

Table 9 shows the detailed statistics of LINT.

### F.1 DOMAIN-WISE HUMAN EVALUATION

As mentioned in § 3.3, we uniformly sample 4,000 statements from LINT for human evaluation. Table 10 provides a more detailed domain-wise statistics on the the data type conformity and factual correctness performance of LINT long-tail knowledge generation. While "Natural Properties" has the highest overall accuracy and factuality, model performance on positive templatesn Table 3 is the lowest while model performance on negative templates is the highest on this domain. This suggests that these LLMs might have been most aligned in this domain during pre-training.

Table 7: Templates used for machine entailment classification task.

|   | Template | Label |
|---|---|---|
| 1 | Is it true that if `premise`, `conclusion`. | Yes |
| 2 | Yes or no: if `premise`, `conclusion`. | Yes |
| 3 | True or false: if `premise`, `conclusion`. | True |
| 4 | Right or Wrong: if `premise`, `conclusion`. | Right |
| 5 | Premise: `premise`. Conclusion: `conclusion`. Does premise entail conclusion? | Yes |
| 6 | Premise: `premise`. Conclusion: `conclusion_negation`. Does premise contradict the conclusion? | Yes |
| 7 | Answer the question with yes or no: if `premise`, `conclusion_question`? | *Depends* |
| 8 | Is it true that if `premise`, `conclusion_negation`. | No |
| 9 | Yes or no: if `premise`, `conclusion_negation`. | No |
| 10 | True or false: if `premise`, `conclusion_negation`. | False |
| 11 | Right or Wrong: if `premise`, `conclusion_negation`. | Wrong |
| 12 | Premise: `premise`. Conclusion: `conclusion_negation`. Does premise entail conclusion? | No |
| 13 | Premise: `premise`. Conclusion: `conclusion`. Does premise contradict the conclusion? | No |

Table 8: A rationale generated by GPT4 with COT shows that even though the model is able to cite the relevant information, it inclines to answer "No" because it cannot "definitively conclude".

| | |
|---|---|
| Input | **Premise**: PersonX is of age 17 years 8 months. **Conclusion**: PersonX cannot consume Bourbon. Does premise entail conclusion? |
| Label | Yes |
| Answer | No |
| Rationale | In many countries, the legal drinking age is 18 or 21. If we do not know the specific country PersonX is in, we cannot definitively conclude that they cannot consume Bourbon. |

Table 9: The number of knowledge statements in different domains in LINT.

| | Temporal | Food and Physical Conditions | Natural Properties | Locational | Total |
|---|---|---|---|---|---|
| Head | 9,385 | 4,524 | 11,864 | 999 | 26,772 |
| Longtail | 9,536 | 4,611 | 11,850 | 1,018 | 27,015 |

Table 10: The factual and data type accuracy of each domain in human evaluation.

| | Data Type | Factuality | Overall |
|---|---|---|---|
| Temporal | 90.18 | 85.27 | 77.38 |
| Food and Physical Conditions | 94.97 | 80.73 | 75.98 |
| Natural Properties | 96.61 | 96.61 | 93.81 |
| Locational | 98.88 | 70.79 | 70.79 |

## F.2 RULE DEFINITIONS

Table 11 shows the definitions of the 6 sampled rules.

Table 11: Rule definitions.

| | |
|---|---|
| Rule0 | lived_in(Person P, Geographic Location A)
& lived_during(Person P, Historical Time Period D)
& existed_during(Geographic Location A, Historical Time Period D)
& was_invented_in(Product or Technology C, Year Y)
& is_more_than_a_century_earlier_than(Historical Time Period D, Year Y)
→ is_not_able_to_use(Person P, Product or Technology C) |
| Rule32 | has_trouble_lifting(Person X, Name of Appliance B)
& is_heavier_than(Object A, Name of Appliance B)
→ cannot_lift(Person X, Object A) |
| Rule46 | is_allergic_to(Person A, Substance X)
& includes(Name of Cosmetics B, Substance X)
→ cannot_use(Person A, Name of Cosmetics B) |
| Rule88 | died_in(Historical Figure A, Historical Time Period X)
& was_created_during(Artifact B, Historical Time Period Y)
& is_earlier_than(Historical Time Period X, Historical Time Period Y)
→ cannot_create(Historical Figure A, Artifact B) |
| Rule112 | has_trouble_containing(Room B, Furniture C)
& is_larger_than(Furniture A, Furniture C)
→ cannot_fit_in(Furniture A, Room B) |
| Rule122 | has_trouble_containing(Trunk B, Furniture C)
& is_larger_than(Furniture A, Furniture C)
→ cannot_fit_in(Furniture A, Trunk B) |

# G AMAZON MECHANIC TURK

## G.1 RECRUITING WORKERS

We use a qualification task to recruit AMT workers. In the qualification task, all workers will be presented with three manually selected statements, which are clear and representative. Each statement has five related questions as described in Appendix B. Only workers who answer all the questions correctly will be recruited. In the end, we recruited 38 workers to evaluate the quality of generation and 17 workers as human baselines for the entailment classification task.

## G.2 TEMPLATES

Figure 10 and Figure 11 show the template we use for the evaluation of generation quality and the entailment classification task.

## G.3 AGREEMENT STATISTICS

Table 12 shows the agreement of annotations in the evaluation task. The high agreement of the data type conformity and factual correctness for LINT ensures the reliability of our results. The agreement for baselines is lower, which also indicates that the generated statements of baselines are of low-quality and confusing for human annotators.

Table 12: Agreement of annotations in the evaluation task.

| Accuracy | ChatGPT | GPT4 | LINK |
|---|---|---|---|
| Data Type | 79.29 | 83.16 | 87.54 |
| Factuality | 38.35 | 58.48 | 75.10 |
| Overall | 65.64 | 74.93 | 83.39 |

## G.4 FAILURE CASE EXAMPLES

We analyze some failure cases that are labeled as incorrect in the human evaluation. Table 13 presents some examples.

Table 13: Examples that are labeled as incorrect during human evaluation. Note that the reasons are analyzed by the authors instead of annotators.

| Rule 21 | Locational | **Rule:** is_located_in(Person A, Location X) & is_forbidden_in(Food Item B, Location X) → cannot_eat(Person A, Food Item B)
**Premise:** Person X is located in Houston
**Conclusion:** Person X cannot eat Chocolate
**Is Houston a location? Annotation:** Yes
**Is Chocolate a food item? Annotation:** Yes
**Does the premise entail the conclusion? Annotation:** No
**Reason:** It is a factual error. Chocolate is not actually forbidden in Houston, so People in Houston can eat chocolate. |
| --- | --- | --- |
| Rule 58 | Food and Physical Conditions | **Rule:** can_treat(Drug B, Name of Disease X) & has(Person A, Name of Disease X) → should_take(Person A, Drug B)
**Premise:** Person X has Hepatitis
**Conclusion:** Person X should take Sofosbuvir
**Is Hepatitis a name of disease? Annotation:** Yes
**Is Sofosbuvir a drug? Annotation:** Yes
**Does the premise entail the conclusion? Annotation:** No
**Reason:** It is a factual error. There are different types of hepatitis viruses. Sofosbuvir is a medication used primarily for the treatment of hepatitis C. For other types of hepatitis, different medications or treatments may be necessary. |
| Rule 74 | Temporal | **Rule:** vanished_in(Plant A, Historical Time Period X) & was_invented_in(Weapon B, Historical Time Period Y) & is_earlier_than(Historical Time Period X, Historical Time Period Y) → cannot_be_used_to_conceal(Plant A, Weapon B)
**Premise:** Plant X vanished in Mongol
**Conclusion:** Plant X cannot be used to conceal M92 Zolja
**Is Mongol a historical time period? Annotation:** No
**Is M92 Zolja a weapon? Annotation:** Yes
**Does the premise entail the conclusion? Annotation:** Yes
**Reason:** It is a data type error. The Mongols are an East Asian ethnic group native to Mongolia, not a time period. The Mongol Empire may refer to a period of the 13th and 14th centuries, but Mongol cannot. |
| Rule 116 | Natural Properties | **Rule:** has_trouble_containing(Drawer B, Tool C) & is_larger_than(Tool A, Tool C) → cannot_be_placed_in(Tool A, Drawer B)
**Premise:** Drawer X has trouble containing Scroll saw
**Conclusion:** Car cannot be placed in Drawer X
**Is Scroll saw a Tool? Annotation:** Yes
**Is Car a Tool? Annotation:** No
**Does the premise entail the conclusion? Annotation:** Yes
**Reason:** It is a data type error. Car is a vehicle instead of a tool. |

**Please read the following Instructions and Examples very carefully, and refer back to them while annotating:**

Instructions (click to expand)

In this HIT you will be presented with a Premise and a Conclusion. The Premise is a fact about a person or object, and the Conclusion is the ability of the person or object (e.g., someone is not able to do something). Your job is to **answer Yes or No to several questions about the Premise and Conclusion**. To answer each question, we request you to **use search engines** to verify your answers.

Below are a few examples:

- Premise: PersonX is allergic to dairy.
  Conclusion: PersonX is not able to eat ice cream.
- Premise: PersonX lived in Vienna and PersonX lived during the Austro-Hungarian Empire.
  Conclusion: PersonX is not able to use solar panels.
- Premise: PersonX was born in Ancient civilizations
  Conclusion: PersonX is able to use Microsoft Teams.

For each Premise and Conclusion, we may ask TWO types of quesitons:

1. **Datatype Correctness**: Does the entities in Premise and Conclusion have right datatepe? (Yes/No)
   - For this question, we ask about the datatype correctness of entities in Premise or Conclusion. The question may consist of several subquestions. **Only if all the subquestions are correct, the answer should be Yes.** Grammatical mistakes and spelling mistakes shall be ignored.
2. **Entailment**: Can the Premise entail the Conclusion? (Yes/No)
   - For this question, we ask about the relationship between Premise and Conclusion. **Yes: it means that the Premise can inevitably lead to the Conclusion. No: it means that given the Premise, the Conclusion may not necessarily be true.** Grammatical mistakes and spelling mistakes shall be ignored.

Example

Premise:

| PersonX lived in Istanbul and PersonX lived during the 12th century. |

Conclusion:

| PersonX is not able to use cloud computing. |

Related to this Premise and Conclusion, we have the following questions:

**Question I.** (1) Is Istanbul a geographic location? (2) Is the 12th century in Premise a historical time period?

**Question II.** Is cloud computing a product or technology?

**Question III.** Does the Premise entail the Conclusion?

Examples (click to expand)

**Ignore grammatical mistakes and spelling mistakes!**
**Please read the following Premise and Conclusion carefully:**

Premise:

| ${premise} |

Conclusion:

| ${conclusion} |

**Now please answer the following questions:**

**I.** ${premise_datatype}

No         Yes

At least one of the entities has wrong datatype.

**II.** ${conclusion_datatype}

No         Yes

At least one of the entities has wrong datatype.

**III.** Does the premise entail conclusion?

No         Yes

The premise cannot entail the conclusion.

(Optional) Please let us know if anything was unclear, if you experienced any issues, or if you have any other feedback for us.

Entry ${entryid}

Submit

Figure 10: AMT template for the evaluation of generation quality.

**Please read the following Instructions and Examples very carefully, and refer back to them while annotating:**

Instructions (click to expand)

We intend this to be a fun task that tests your ability to reason on factual knowledge. We pay a base of $0.03 per annotation. Bonus is calculated as 2 tiers:

1. **Doing great!** - if your overall accuracy > 90% or made no more than 1 mistakes per batch (whichever criteria is more relaxed), we bonus you $0.08 per annotation.
2. **Way to go!** - if your overall accuracy is lower than 90% or made more than 2 mistakes per batch, we will compensate you with a flat rate of participation reward.

In this HIT you will be presented with a Sentence composed of a Premise and a Conclusion. The Premise is a fact about a person or object, and the Conclusion is the ability of the person or object (e.g., someone is not able to do something). Your job is to **answer the binary choice question in the Sentence**. **Grammatical mistakes and spelling mistakes shall be ignored**.

Below are a few examples of Sentences:

- Yes or no: if Person X was born in The Renaissance, Person X was able to use Microsoft Word.
- Yes or no: if Person X is allergic to Fish, Person X is able to eat Sushi.
- Yes or no: if Person X lived in Ireland and Person X lived during Napoleonic Era, Person X is not able to use Corel Draw.

**You are required to use search engines to verify your answers.** Do NOT use AI Chatbot interfaces such as ChatGPT. *If you do not follow instructions, your work may face rejection.*

If you have questions and want to email us, please include your worker id and HIT ID and title.

Examples (click to expand)

**Ignore grammatical mistakes and spelling mistakes!**
**Please read the following Sentence carefully:**

Sentence:

| ${sentence} |

**Now please answer the question following the prompt instruction in the above Sentence:**

${negative}         ${positive}

The answer should be No/False/Wrong.

(Optional) Please let us know if anything was unclear, if you experienced any issues, or if you have any other feedback for us.

Entry ${entryid}

Submit

Figure 11: AMT template for the human baseline of the entailment classification task.

