# OpenReview forum: "In Search of the Long-Tail: Systematic Generation of Long-Tail Knowledge via Logical Rule Induced Search"
_ICLR.cc/2024/Conference — ICLR 2024 Conference Withdrawn Submission_

### Official Review · Reviewer_tVDU · 2023-10-30

**Soundness:** 2 fair
**Presentation:** 3 good
**Contribution:** 2 fair
**Rating:** 5
**Confidence:** 4

**Summary:**

The paper attempts to evaluate LLMs from a new perspective: long-tail prompts. To this end, the authors craft a few symbolic rules, and instantiate the rules with a knowledge model, a critic, and a re-ranker. By this means, they build a long-tail dataset and compare InstructGPT, ChatGPT, and GPT-4 on the dataset (as well as with some head prompts).  By making the dataset open-sourced, the authors could facilitate the research of LLMs in terms of inability.

While the starting point of the paper is interesting, I have several concerns:

(1) the gaps between the long tail and the head are small: the main conclusion of the work is that existing LLMs fall short on long-tail data. However, I am not fully convinced if the conclusion really holds since the performance drop, as shown in Table 3, only 1%~3%. I am not saying the results mean nothing, as the drop is consistent across all models, but I am expecting more insightful analysis. For example, how the models perform over different types of rules. Is it possible that a model performs much worse on one type of rules than others and thus results in the overall drop? Is "long-tail (or in other words, infrequent knowledge)" the only reason that leads to the performance drop? Is it possible that because advanced models are well aligned and denied to return definite answers more frequently on the long-tails and thus they perform badly?

(2) I am not convinced by the claim "GPT-4 cannot generate long-tail data", since LINK also leverages LLMs like ChatGPT and GPT-4 in data generation. It seems that the difference comes from the exploitation of the critic and the re-ranker. Then how about applying the same critic and the re-ranker to the data generated in Section 3?

(3) Why no results on open-sourced LLMs?

(4) Can the authors provide more details on how the meta rules are crafted? Are there any guidelines or principles behind?

**Strengths:**

a new resource for research of LLMs

**Weaknesses:**

inadequate analysis to support the main conclusion
no results on open-sourced LLMs

**Questions:**

Please refer to the summary.

---

> ### Author Response · Authors · 2023-11-19
> **Response to Reviewer tVDU (Part 1)**
>
> Thank you for your suggestions and questions! Below are our responses to each of them.
>
> **Q1.** the gaps between the long tail and the head are small: the main conclusion of the work is that existing LLMs fall short on long-tail data. However, I am not fully convinced if the conclusion really holds since the performance drop, as shown in Table 3, only 1%~3%. I am not saying the results mean nothing, as the drop is consistent across all models, but I am expecting more insightful analysis. For example, **(a)** how the models perform over different types of rules. Is it possible that a model performs much worse on one type of rules than others and thus results in the overall drop? Is "long-tail (or in other words, infrequent knowledge)" the only reason that leads to the performance drop? **(b)** Is it possible that because advanced models are well aligned and denied to return definite answers more frequently on the long-tails and thus they perform badly?
>
> (a) Thank you for raising this question! To further explain our result, we conducted a more detailed analysis and updated Table 3 and Section 4. We conducted separate analysis on the four domains, and provided more insights on the discrepancy between the positive and negative template performance.
>
> First, we found that for negative templates the long-tail distribution drops ubiquitously for every model and every domain, the highest drop being as large as 13%. The positive template performance increases for most domains but one, which may be the reason why the overall performance does not drop significantly for longtail distribution. As for human performance baseline, we do not see a statistically significant drop for the long-tail distribution. It is expected because humans can verify their answers with search engines, and their reasoning process does not interfere with the infrequent knowledge. We also included stats on the number of data we have for each domain in Appendix F Table 9.
>
> (b) The example we used to demonstrate that LLMs deny to return “definite answers” happen mostly in the positive template evaluations, i.e. where the expected answer is yes but the model returns no. In the long-tail distribution, the performance on positive templates in fact increased, compared to the head distribution. This shows that when facing less frequent knowledge, LLMs are less “aligned" with its training guidance of “not returning definite answers”.
>
> **Q2.** I am not convinced by the claim "GPT-4 cannot generate long-tail data", since LINK also leverages LLMs like ChatGPT and GPT-4 in data generation. It seems that the difference comes from the exploitation of the critic and the re-ranker. Then how about applying the same critic and the re-ranker to the data generated in Section 3?
>
> Thank you to your suggestion! Simply applying critic and reranker on GPT4 generated results cannot achieve the same effect as step-wise filtering and reranking. In Appendix D.2, we post-hoc apply our reranker on the search results from GPT4 using baseline prompting methods, and present the distribution plots in Figure 8. The distribution after post-hoc reranker still cannot achieve separation between head and longtail. This shows that applying the reranker at each step of the search is absolutely essential.
>
> **Q3.** Why no results on open-sourced LLMs?
>
> Given that generating longtail data from symbolic rules is a difficult task, we decided to test it on the best models out there, which are the OpenAI GPT-3+ models. We will definitely test both the generative and entailment task on open source LLMs in later versions.

---

> > ### Author Response · Authors · 2023-11-19
> > **Response to Reviewer tVDU (Part 2)**
> >
> > **Q4**. Can the authors provide more details on how the meta rules are crafted? Are there any guidelines or principles behind?
> >
> > We create the meta rules with the principles stated in Section 2.2. Here are some general principles and a more detailed guideline copied from Section 2.2:
> >
> > **The symbolic rule should have a subject.**
> >
> > 1. The symbolic rule should be topically referring to a person or an object.
> >
> > **The symbolic rule should have some degree of complexity.**
> >
> > 2. The premise and conclusion should not be paraphrases.
> > 3. The symbolic rule should contain at least 3 variables and 2 predicates in the premise.
> >
> > **The sybolic rule should be searchable and verifiable.**
> >
> > 4. The symbolic rule should be linearly chained. This is to make sure be can find dependency for every variable, and the rule does not include distractors.
> > 5. The symbolic rule should not contain predicates that are potentially out of scope of LLMs’ knowledge.

---

> > > ### Author Response · Authors · 2023-11-22
> > > **Friendly reminder to respond to author rebuttal**
> > >
> > > Dear Reviewer tVDU,
> > >
> > > Thank you again for your review! We are happy to hear that you appreciate LINK’s contribution and presentation. Based on your thoughtful feedback, we wrote a detailed rebuttal covering the following points:
> > >
> > > 1. Expanded detailed analysis on the performance difference between head and longtail distribution in Section 4.
> > > 2. Addition of a series of ablation studies **on the effect of critic and reranker**  in Appendix D.
> > > 3. Discussion on model alignment with model behaviors on positive and negative templates.
> > > 4. Discussion on the principles of creating the meta rules.
> > > 5. Discussion on the reasons for our model choices.
> > >
> > > We would love to hear your thoughts about our rebuttal, including whether it sufficiently addresses your concerns and questions. If you believe that our rebuttal is satisfactory, it would be great if you could consider increasing your score. Any feedback is welcome and greatly appreciated!
> > >
> > > Sincerely,
> > >
> > > Paper 6959 Authors

---

> > > > ### Author Response · Authors · 2023-11-23
> > > > **Another friendly reminder to respond to author rebuttal**
> > > >
> > > > Dear Reviewer tVDU,
> > > >
> > > > Thank you again for your review! We have revised our paper based on your suggestions and questions.
> > > >
> > > > We would love to hear your thoughts about our rebuttal, including whether it sufficiently addresses your concerns and questions. If you believe that our rebuttal is satisfactory, it would be great if you could consider increasing your score. Any feedback is welcome and greatly appreciated!
> > > >
> > > > Sincerely,
> > > >
> > > > Paper 6959 Authors

---

### Official Review · Reviewer_zFAF · 2023-10-31

**Soundness:** 2 fair
**Presentation:** 3 good
**Contribution:** 2 fair
**Rating:** 3
**Confidence:** 3

**Summary:**

This paper addresses the challenge of evaluating large language models (LLMs) by generating long-tail data, which is inherently difficult for these models. To tackle this, the authors introduce the Logic-Induced-Knowledge-Search (LINK) framework, which employs symbolic rules to prompt LLMs to generate long-tail knowledge statements. These statements are then assessed for correctness by a critic model and sorted into long-tail or head distribution categories using a reranker. The resulting dataset, Logic-Induced-Long-Tail (LINT), is found to contain 89% factually correct statements. In contrast, state-of-the-art LLMs like ChatGPT and GPT4 struggle to generate accurate long-tail statements. To verify if long-tail data are more challenging for LLM, they conducted experiments on entailment classification tasks and found that the performance dropping by 2% to 4% when dealing with long-tail knowledge. This work highlights the effectiveness of LINK in generating long-tail data and emphasizes its importance in evaluating LLMs in challenging scenarios.

**Strengths:**

They proposed a framework to generate long-tail data that are more challenging for LLM and the experiments verified that LLM performs worse on the long-tail data compared with in the head distribution.

**Weaknesses:**

* The paper demonstrates that their long-tail data generation framework outperforms LLMs like ChatGPT and GPT4. However, this comparison might not be entirely fair. They've introduced a critic model to refine the generated statements and a reranker to prioritize them, which naturally could boost performance. The authors should provide results from an ablation study where the critic model and reranker are removed. The improvements in Table 2 are likely due to the critic model's role in filtering statements.
* Their generated statements are reorganized based on likelihood, which is the same metric used to evaluate data quality in Figure 3 and Figure 4. Without applying the same reranker to ChatGPT and GPT4's generated results, a direct comparison may not be equitable. It's not surprising that LINK's data appears well-separated because they've already been ranked using the same metric, although by a different model.
* The practical application of this framework appears limited. It's not straightforward to use this approach for generating long-tail data in various other tasks. The logic-induced reasoning and generated statements may not have broad applicability beyond their specific context.
* The observed performance drop in LLMs between head distribution and long-tail data is relatively small, around 2% to 4%. This raises questions about the significance of this work.

**Questions:**

N/A

---

> ### Author Response · Authors · 2023-11-19
> **Response to Reviewer zFAF (Part 1)**
>
> Thank you for your suggestions and questions! Below are our responses to each of them.
>
> **Q1.** The paper demonstrates that their long-tail data generation framework outperforms LLMs like ChatGPT and GPT4. However, this comparison might not be entirely fair. They've introduced a critic model to refine the generated statements and a reranker to prioritize them, which naturally could boost performance. The authors should provide results from an ablation study where the critic model and reranker are removed. The improvements in Table 2 are likely due to the critic model's role in filtering statements.
>
> Thank you for raising this question! We acknowledge that adding a critic and reranker in LINK helps the performance. However, removing the critic and reranker model means that foreseeably InstructGPT, a much weaker model, may not generate as well as stronger models like GPT4. As the time constraint for the rebuttal does not allow us to do extensive human evaluation on accuracy, we added an ablation study on removing the reranker. We generated knowledge for a few rules by replacing the reranker with a random sampling method (Appendix D.1, Figure 7). When we remove reranker from LINK, the distribution of the resulting head and long-tail statements are pulled towards the center, so it shows that a reranker is essential for separating the head and long-tail distribution.
>
> Nonetheless, we agree with your suggestion that we should conduct ablation studies on the critic and reranker, so **instead we added critic and reranker to GPT4**, and compared it with baseline prompting method:
>
> We used GPT4 as the knowledge model in LINK, in combination with the same critic and reranker (Appendix D.3). Due to time constraint for the rebuttal, we only tested on one rule (rule 88). We conducted human evaluation on the generations and compared it against the GPT4 baseline prompting generations and generations using LINK with InstructGPT (Table 6). For this particular rule, using GPT4 improves both data type and factuality correctness from using InstructGPT. GPT4 with LINK also significantly improves for factuality and overall accuracy from GPT4 with baseline prompting, raising the overall accuracy from 66% to 79%. We plotted the distribution of the statements (Figure 9), and the plot shows that GPT4 with LINK generates statements that fall in a similar distribution with InstructGPT with LINK, while GPT4 with baseline prompting cannot produce separable longtail and head distribution statements. This result provides stronger evidence that in combination with two smaller open source models as critic and reranker, we can get better search results using a weaker model and a smaller budget than stronger models.
>
> **Q2.** Their generated statements are reorganized based on likelihood, which is the same metric used to evaluate data quality in Figure 3 and Figure 4. Without applying the same reranker to ChatGPT and GPT4's generated results, a direct comparison may not be equitable. It's not surprising that LINK's data appears well-separated because they've already been ranked using the same metric, although by a different model.
>
> We acknowledge that likelihood is both used during search and evaluation (thought with a different model), but we want to emphasize a few key points:
>
> 1) Simply applying critic and reranker on GPT4 generated results cannot achieve the same effect as step-wise filtering and reranking. In Appendix D.2, we post-hoc apply our reranker on the search results from GPT4 using baseline prompting methods, and present the distribution plots in Figure 8. The distribution after post-hoc reranker still cannot achieve separation between head and longtail. This shows that applying the reranker at each step of the search is absolutely essential.
>
> 2) The statements on which we calculate likelihood during search are different from the statements whose distribution we evaluate. The exact statements used at each step of the knowledge beam search is different. The statements that are used to rank the knowledge beams are shorter (only consisting of the already-searched predicates), but they achieve the separation that post-hoc filtering cannot achieve.

---

> ### Author Response · Authors · 2023-11-19
> **Response to Reviewer zFAF (Part 2)**
>
> **Q3.** The practical application of this framework appears limited. It's not straightforward to use this approach for generating long-tail data in various other tasks. The logic-induced reasoning and generated statements may not have broad applicability beyond their specific context.
>
> One application of our framework is for data augmentation and creating more challenging evaluation data. One can take natural language statements from existing benchmarks (or new benchmark proposals) and turn them into symbolic rules. From these symbolic rules we can generate unseen and longtail data using LINK.This could be verify useful for augmenting data and mitigating benchmark leakage during evaluation.
>
> **Q4.** The observed performance drop in LLMs between head distribution and long-tail data is relatively small, around 2% to 4%. This raises questions about the significance of this work.
>
> Thank you for raising this question! To further explain our result, we conducted a more detailed analysis and updated Table 3 and Section 4. We conducted separate analysis on the four domains, and provided more insights on the discrepancy between the positive and negative template performance.
>
> First, we found that for negative templates the long-tail distribution drops ubiquitously for every model and every domain, the highest drop being as large as 13%. The positive template performance increases for most domains but one, which may be the reason why the overall performance does not drop significantly for longtail distribution. As for human performance baseline, we do not see a statistically significant drop for the long-tail distribution. It is expected because humans can verify their answers with search engines, and their reasoning process does not interfere with the infrequent knowledge. We also included stats on the number of data we have for each domain in Appendix F Table 9.

---

> > ### Author Response · Authors · 2023-11-22
> > **Friendly reminder to respond to author rebuttal**
> >
> > Dear Reviewer zFAF,
> >
> > Thank you again for your review! We are happy to hear that you appreciate LINK’s contribution and presentation. Based on your thoughtful feedback, we wrote a detailed rebuttal covering the following points:
> >
> > 1. Expanded detailed analysis on the performance difference between head and longtail distribution in Section 4.
> > 2. Addition of a series of ablation studies **on the effect of critic and reranker**  in Appendix D.
> > 3. Ablation study on post-hoc rerankers to show that reranking the beams **at each step of search** is the key reason for improving LINK search results.
> > 4. Discussion on future work and practical applications of our work.
> >
> > We would love to hear your thoughts about our rebuttal, including whether it sufficiently addresses your concerns and questions. If you believe that our rebuttal is satisfactory, it would be great if you could consider increasing your score. Any feedback is welcome and greatly appreciated!
> >
> > Sincerely,
> >
> > Paper 6959 Authors

---

> > > ### Author Response · Authors · 2023-11-23
> > > **Another friendly reminder to respond to author rebuttal**
> > >
> > > Dear Reviewer zFAF,
> > >
> > > Thank you again for your review! We have revised our paper based on your suggestions and questions.
> > >
> > > We would love to hear your thoughts about our rebuttal, including whether it sufficiently addresses your concerns and questions. If you believe that our rebuttal is satisfactory, it would be great if you could consider increasing your score. Any feedback is welcome and greatly appreciated!
> > >
> > > Sincerely,
> > >
> > > Paper 6959 Authors

---

### Official Review · Reviewer_TKVD · 2023-11-02

**Soundness:** 2 fair
**Presentation:** 2 fair
**Contribution:** 2 fair
**Rating:** 3
**Confidence:** 4

**Summary:**

This paper aims to systematically find failure cases of large language models (LLMs), specifically, long-tail statements, i.e., the statements with lower likelihood. The authors propose a Logic-Induced-Knowledge-Search (LINK) framework to generate long-tail knowledge statements, where a pipeline is designed to prompt LLMs for generating statements with low likelihood. Based on 14 human-written meta-rules, the proposed LINK approach constructs a dataset with 200 other rules and 40,000 long-tail knowledge statements, in the (premise, conclusion) format.

**Strengths:**

I don't see clear strengths in this paper.

**Weaknesses:**

*  Although this paper claims the proposed approach to be "systematical", I still have a concern on its scalability. All the 40,000 outcome statements are actually based on 14 meta-rules come up by human. Scalably obtaining such meta rules can be challenging, and still rely on human curation.
* The 200 symbolic rules in the constructed dataset actually come from slightly changing some wording from the 14 meta rules, which raises my concern on the quality of the dataset, especially about the diversity among the outcome knowledge statements.
*  There are several other similar resources, such as ATOMIC with 1.3M knowledge statements, but this paper doesn't discuss and compare, besides mentioning they have similar data format.
* The experiments in this paper don't have a solid and convincing conclusion. For example, in the proposed LLM evaluation with the constructed dataset, in the entailment classification task, the LLM performances drop by 2 - 4% on the long-tail statements, compared with the ones from head distributions. However, from Table 3, the human annotated accuracy also drops by 2%. It is unclear what this evaluation suggests.
* As a dataset, only 89% of the statements are annotated factually correct, which is not satisfying. Besides, there is no analysis conducted on the 10% failure cases, where the proposed framework produces factual errors based on correct logic rules.

[1] Hwang, Jena D. et al. “COMET-ATOMIC 2020: On Symbolic and Neural Commonsense Knowledge Graphs.” AAAI 2020.

**Questions:**

See weaknesses.

---

> ### Author Response · Authors · 2023-11-19
> **Response to Reviewer TKVD (Part 1)**
>
> Thank you for your suggestions and questions! Below are our responses to each of them.
>
> **Q1.** All the 40,000 outcome statements are actually based on 14 meta-rules come up by human. Scalably obtaining such meta rules can be challenging, and still rely on human curation. The 200 symbolic rules in the constructed dataset actually come from slightly changing some wording from the 14 meta rules, which raises my concern on the quality of the dataset, especially about the diversity among the outcome knowledge statements.
>
> It is true that one meta-rule can only produce one form of statements, it is also important to note that given one meta-rule, we can already scale it up to a large number of statements. To increase the diversity of the outcome knowledge statements, one may scale up the meta rule by leveraging the powerful inferential knowledge and generative capabilities of LLMs.
>
> For example, one may utilize the premise and conclusion in the existing meta rules as examples to prompt GPT4 to generate corresponding feasible premises and conclusions, thereby constructing more symbolic rules. Moreover, one may randomly select two abstract objects to generate potential relationships between them to form the conclusions in the rules, from which we can produce more premises using the same prompting method for GPT4. These machine-generated symbolic rules can go through a combination of model-based and human-based verification and become the foundation for generating more diverse knowledge statements using LINK.
>
> **Q2.** There are several other similar resources, such as ATOMIC with 1.3M knowledge statements, but this paper doesn't discuss and compare, besides mentioning they have similar data format.
>
> ATOMIC (Sap, 2019) was created with a different purpose than LINT: ATOMIC is a knowledge graph and LINT is a dataset for longtail knowledge statements; ATOMIC is evaluated on its coverage while LINT is evaluated on its data distribution. Even though both papers evaluate model’s ability to generate knowledge, ATOMIC evaluates models on a conditional sequence generation problem while LINK evaluates models with a value generation problem.
>
> The follow-up work, COMET-ATOMIC (Hwang, 2020) focuses on expanding the coverage of knowledge graphs from ATOMIC 2019, and also studies whether training models on knowledge graphs will make it a better knowledge model. Thus directly comparing our work with both ATOMIC works is difficult, provided that our goals and methodologies are different.
>
> [1] Sap, M., Le Bras, R., Allaway, E., Bhagavatula, C., Lourie, N., Rashkin, H., ... & Choi, Y. (2019, July). Atomic: An atlas of machine commonsense for if-then reasoning. In Proceedings of the AAAI conference on artificial intelligence (Vol. 33, No. 01, pp. 3027-3035).
>
> [2] Hwang, J. D., Bhagavatula, C., Le Bras, R., Da, J., Sakaguchi, K., Bosselut, A., & Choi, Y. (2021, May). (Comet-) atomic 2020: on symbolic and neural commonsense knowledge graphs. In Proceedings of the AAAI Conference on Artificial Intelligence (Vol. 35, No. 7, pp. 6384-6392).

---

> ### Author Response · Authors · 2023-11-19
> **Response to Reviewer TKVD (Part 2)**
>
> **Q3.** The experiments in this paper don't have a solid and convincing conclusion. For example, in the proposed LLM evaluation with the constructed dataset, in the entailment classification task, the LLM performances drop by 2 - 4% on the long-tail statements, compared with the ones from head distributions. However, from Table 3, the human annotated accuracy also drops by 2%. It is unclear what this evaluation suggests.
>
> Thank you for raising this question! To further explain our result, we conducted a more detailed analysis and updated Table 3 and Section 4. We conducted separate analysis on the four domains, and provided more insights on the discrepancy between the positive and negative template performance.
>
> First, we found that for negative templates the long-tail distribution drops ubiquitously for every model and every domain, the highest drop being as large as 13%. The positive template performance increases for most domains but one, which may be the reason why the overall performance does not drop significantly for longtail distribution. As for human performance baseline, we do not see a statistically significant drop for the long-tail distribution. It is expected because humans can verify their answers with search engines, and their reasoning process does not interfere with the infrequent knowledge. We also included stats on the number of data we have for each domain in Appendix F Table 9.
>
> **Q4.** As a dataset, only 89% of the statements are annotated factually correct, which is not satisfying. Besides, there is no analysis conducted on the 10% failure cases, where the proposed framework produces factual errors based on correct logic rules.
>
> Thank you for this question! We added an analysis on human evaluation accuracy for each domain in Appendix F Table 10. We found that while “Natural Properties” has the highest overall accuracy and factuality, model performance on positive templates in Table 3 is the lowest while model performance on negative templates is the highest on this domain. This trend is opposite to the other domains. This suggests that LLMs might have been most aligned in this domain during pre-training. We also included some examples of failure cases in Appendix G.4, Table 13.

---

> > ### Author Response · Authors · 2023-11-22
> > **Friendly reminder to respond to author rebuttal**
> >
> > Dear Reviewer TKVD,
> >
> > Thank you again for your review! Based on your thoughtful feedback, we wrote a detailed rebuttal covering the following points:
> >
> > 1. Expanded detailed analysis on the performance difference between head and longtail distribution in Section 4..
> > 2. Addition of a series of ablation studies **on the effect of critic and reranker**  in Appendix D.
> > 3. Domain-specific information on the errors that LINK made according to human evaluation in Appendix F.1 and G.4.
> > 4. Discussed the **difference** between our work and other works with **similar dataset format**.
> > 5. Discussed the scalability of the dataset through meta rules.
> >
> > We would love to hear your thoughts about our rebuttal, including whether it sufficiently addresses your concerns and questions. If you believe that our rebuttal is satisfactory, it would be great if you could consider increasing your score. Any feedback is welcome and greatly appreciated!
> >
> > Sincerely,
> >
> > Paper 6959 Authors

---

> > > ### Author Response · Authors · 2023-11-23
> > > **Another friendly reminder to respond to author rebuttal**
> > >
> > > Dear Reviewer TKVD,
> > >
> > > Thank you again for your review! We have revised our paper based on your suggestions and questions.
> > >
> > > We would love to hear your thoughts about our rebuttal, including whether it sufficiently addresses your concerns and questions. If you believe that our rebuttal is satisfactory, it would be great if you could consider increasing your score. Any feedback is welcome and greatly appreciated!
> > >
> > > Sincerely,
> > >
> > > Paper 6959 Authors

---

### Official Review · Reviewer_pztA · 2023-11-05

**Soundness:** 3 good
**Presentation:** 4 excellent
**Contribution:** 4 excellent
**Rating:** 8
**Confidence:** 4

**Summary:**

The paper presents a sampling strategy and algorithm (Logic-Induced-Knowledge-Search, LINK) to systematically generate long-tail knowledge statements, as determined by a reference large language model (LLM).  A core component of the sampling strategy is to leverage algorithmically generated symbolic logic rules as prompt templates, which are then used to sample a target LLM for potential long-tail variable values.  The candidate samples thus generated are then verified for accuracy by a critic model and ensured to fit the long-tail distribution by a re-ranker. The authors create a dataset called Logic-Induced-Long-Tail (LINT) containing 200 symbolic rules and 40,000 knowledge statements across four domains, with human annotations confirming 89% factual accuracy. This method surpasses traditional direct generation of long-tail samples from LLMs, where two tested SOTA models achieved only 61% and 79% accuracy in creating factually correct long-tail statements when using only self-knowledge. This process is given the moniker knowledge beam search, and involves sequentially prompting for variable values, validating data type and factual correctness, and then re-ranking to maintain the focus on long-tail information.

**Strengths:**

Originality:  The data set and reference code used to create it is novel, as is the search strategy.

Quality:  The sampling strategy, as outlined, is technically correct and feasible, and performance comparisons are given between different models.  Evaluations with and without LINK are discussed.  The dataset (LINT) appears to be well constructed.

Clarity:  The narrative discussion is clear.

Significance:  Techniques such as this should be useful for systematic importance sampling from LLMs, providing new ways to quantify the performance of LLMs.

**Weaknesses:**

Quality:  (1) A more statistically rigorous discussion of long tail statistics and how that effects LINK's performance would help highlight the importance of this approach.  (2) a descriptive discussion of the LINT dataset would be enlightening.

Clarity: Visualizations of more extensive evaluation of long tail statistics would be helpful.

**Questions:**

This is a very nice idea and well written paper.

1. Do you have a grounding for the statement "it has become increasingly harder for researchers to find tasks that are still challenging to the models"?  The assertion does not seem relevant / central to the work.

2. Could you expand on your discussion of bias?  What was the specific methodology to study the bias of your sampling method?  Did any of the human feedback suggest an unforeseen bias in the method?

3. Can you present a more rigorous discussion of the distribution approximation properties for the method (although probably in a simpler context)?

4. What future work do you foresee in this area?

---

> ### Author Response · Authors · 2023-11-19
> **Response to Reviewer pztA (Part 1)**
>
> Thank you for your suggestions and questions! Below are our responses to each of them.
>
> **Q1.** A more statistically rigorous discussion of long tail statistics and how that effects LINK's performance would help highlight the importance of this approach.
>
> We conducted a series of ablation studies on LINK’s components, including:
>
> 1) We conducted a more detailed analysis and updated Table 3 and Section 4. We conducted separate analysis on the four domains, and provided more insights on the discrepancy between the positive and negative template performance. (See answer to Reviewer TKVD Q3)
>
> 2) We added an ablation study on removing the reranker in LINK. We generated knowledge for a few rules by replacing the reranker with a random sampling method (Appendix D.1, Figure 7) (See answer to Reviewer zFAF Q1).
>
> 3) We added critic and reranker to GPT4, and compared it with baseline prompting method for one rule (Appendix D.3) (See answer to Reviewer zFAF Q1).
>
> 4) In Appendix D.2, we post-hoc apply our reranker on the search results from GPT4 using baseline prompting methods, and present the distribution plots in Figure 8 (See answer to Reviewer zFAF Q2).
>
> **Q2.** A descriptive discussion of the LINT dataset would be enlightening.
>
> We added some statistics about the four domains of the LINT dataset in Appendix F Table 9. Is there a specific aspect of LINT that you would like us to elaborate?
>
> **Q3.** Visualizations of more extensive evaluation of long tail statistics would be helpful.
>
> We included additional distribution plots comparing LINK, ChatGPT and GPT4 generations of more symbolic rules in Appendix C.1 Figure 5. We also included the factual and data type accuracy of long-tail generations in each domain in human evaluation (Appendix F.1 Table 10, see answer to Reviewer TKVD Q4).
>
> **Q4.** Do you have a grounding for the statement "it has become increasingly harder for researchers to find tasks that are still challenging to the models"? The assertion does not seem relevant / central to the work.
>
> The paper “Measuring progress on scalable oversight for large language models” highlights the challenge of supervising systems that have knowledge or capabilities we lack as a “typical human”, where most researchers are “typical humans” on most tasks compared to “experts”. We will update the abstract with “it has become increasingly harder to find challenging examples from existing benchmarks” after the rebuttal period (due to the updated instruction to not update the abstract during rebuttal).
>
> [1] Bowman, S. R., Hyun, J., Perez, E., Chen, E., Pettit, C., Heiner, S., ... & Kaplan, J. (2022). Measuring progress on scalable oversight for large language models. arXiv preprint arXiv:2211.03540.

---

> ### Author Response · Authors · 2023-11-19
> **Response to Reviewer pztA (Part 2)**
>
> **Q5.** Could you expand on your discussion of bias? What was the specific methodology to study the bias of your sampling method? Did any of the human feedback suggest an unforeseen bias in the method?
>
> One source of sampling bias might come from using a smaller model to verify and rerank the generated knowledge. Given that Flan-T5 is much smaller than InstructGPT, it might create more false negatives during its verification if we take argmax between “yes” and “no” tokens. To account for that, we applied dynamic threshold (Appendix B), where we would dynamically decrease the threshold of the yes logit to a point where we may still accept the correct values despite the low model confidence. A similar concern also exists when we use llama-7B to rerank the knowledge beams to approximate InstructGPT log likelihood. We plot the head and longtail statements distribution over InstructGPT and llama-7B in a scatter plot (Appendix C.2) and show that the statements reranked by llama-7B are sufficiently separable by InstructGPT. So far we have not discovered any unforeseen bias in the sampling method based on human feedback.
>
> **Q6.** Can you present a more rigorous discussion of the distribution approximation properties for the method (although probably in a simpler context)?
>
> We used llama-7B to rerank the partial knowledge statements during knowledge search. One may be concerned that a smaller model cannot well approximate a larger model’s distribution well. We plot the head and longtail statements distribution over InstructGPT and llama-7B in a scatter plot (Appendix C.2) and show that the statements reranked by llama-7B are sufficiently falling in a similar distribution for InstructGPT. Analogously, using InstructGPT to approximate the true longtail distribution of an oracle model can also cause a similar range of error, but this is the currently largest model with log likelihood available. Given that GPT4 is opening up the log likelihood soon, we can reproduce this analysis with GPT4 log likelihood.
>
> **Q7.** What future work do you foresee in this area?
>
> One can build an inductive learning component that precedes LINK to generate long-tail statements. Given this component that can generalize a few examples into symbolic rules, we can apply this framework + LINK on existing benchmarks (or new benchmark proposals) to unseen and longtail statements. This could be verify useful for augmenting data and mitigating benchmark leakage during evaluation.

---

> ### Author Response · Authors · 2023-11-22
> **Friendly reminder to respond to author rebuttal**
>
> Dear Reviewer pztA,
>
> Thank you again for your review! We are happy to hear that you appreciate LINK’s contribution, motivation, and presentation. Based on your thoughtful feedback, we wrote a detailed rebuttal covering the following points:
>
> 1. Expanded detailed analysis on the performance difference between head and longtail distribution in Section 4.
> 2. Addition of a series of ablation studies **on the effect of critic and reranker**  in Appendix D.
> 3. Domain-specific information on the errors that LINK made according to human evaluation in Appendix F.1 and G.4.
> 4. Discussion on our **distribution approximation method**.
> 5. Discussion on future work and practical applications of our work.
> 6. Discussion on potential sampling bias of our method.
>
> We would love to hear your thoughts about our rebuttal, including whether it sufficiently addresses your concerns and questions. If you believe that our rebuttal is satisfactory, it would be great if you could consider increasing your score. Any feedback is welcome and greatly appreciated!
>
> Sincerely,
>
> Paper 6959 Authors

---

> ### Comment · Reviewer_pztA · 2023-11-23
>
> I thank the authors for your thoughtful consideration of the questions posed.  Your explanations improve the presentation, and the additional insights are appreciated.  On the topic of "rigorous discussion of the distribution approximation properties", I was hopeful you had a simple conceptual (back-of-the-envelope) sampling model to provide further insight.  Your answer was sufficient.
>
> I maintain my position that this is a good work on an important topic, and retain my rating of 8 -- accept, good paper.

---

### Author Response · Authors · 2023-11-19
**Message to all Reviewers**

We thank all of the reviewers for their thoughtful feedback and recognition of our paper’s contributions!

First of all, we want to notify you that we have uploaded version 2 of our dataset on https://zenodo.org/records/10126935 and also included ChatGPT and GPT4’s baseline prompting generations. In this version, we slightly improved both LINK’s algorithm and the baseline algorithm. We minimally modified the beam knowledge search algorithm in LINK and diversified the knowledge beams. We also revised our deduplicate method for baseline prompting generations and improved generation quality. All conclusions in the original paper still hold in the rebuttal version. All analyses done in the rebuttal version are based on the version 2 of our dataset.

In response to the reviewers’ comments/questions, we have addressed the following items in our rebuttal and updated paper:

1. We expanded on the results and analysis in Section 4 and Table 3 to explain why despite the current similarity of model performance on entailment classification task between head and longtail distribution, our data **LINT is still effective for evaluating LLMs**. In addition to including **domain specific performance** in the table, we also discussed the different effect longtail data has on LLMs on **positive templates** and **negative templates**. (**pztA**, **TKVD**, **zFAF**, **tVDU**)
2. We highlighted the significance of the LINK framework by adding a series of ablation studies **on the effect of critic and reranker**. These analyses can be found in Appendix D. (**pztA**, **TKVD**, **zFAF**, **tVDU**)
3. We included failure cases of each domain during human evaluation in Appendix G. (**TKVD**)
4. We provided more domain-specific information on the factual errors that LINK made according to human evaluation. (**pztA**, **TKVD**)
5. We provided more insights on our **distribution approximation method** using stepwise reranker in Appendix C. (**pztA**)
6. We pointed out that reranking the statements for LINK is not the key reason that LINT data can be well separated, but rather reranking the beams **at each step of search** is the key reason, accompanied by ablation studies on post-hoc applying rerankers onto model search results. (**zFAF**)
7. We discussed future work and practical applications of our work. (**pztA**, **zFAF**)
8. We discussed the difference between our work and other works with similar dataset format. (**TKVD**)
9. We discussed the potential sampling bias of our method. (**pztA**)
10. We discussed the scalability of the dataset through meta rules. (**TKVD**)
11. We discussed the principles of creating the meta rules. (**tVDU**)
12. We discussed the reasons for our model choices. (**tVDU**)

---

### Author Response · Authors · 2023-11-22
**Message to Reviewers and S/ACs**

Dear ICLR chairs,

Thank you for your efforts in organizing this peer-review process! The reviews we received on our paper were generally quite helpful. In particular, we were happy to find that Reviewer **pztA** appreciates our originality, quality, clarity and significance of contribution, while also giving many suggestions on how to improve the paper. In addition, we appreciate that other reviewers recognize the originality of our work.

In response to the reviewers’ comments/questions, we have addressed the following items in our rebuttal and updated paper:

1. We included more details showing why **LINT is effective for evaluating LLMs**. We expanded on the results and analysis in Section 4 and Table 3 with **domain specific performance** on entailment classification task between head and longtail distribution. We also discussed the different effect longtail data has on LLMs on **positive templates** and **negative templates**. (**pztA**, **TKVD**, **zFAF**, **tVDU**)
2. We highlighted the significance of the LINK framework by adding a series of ablation studies **on the effect of critic and reranker**. These analyses can be found in Appendix D. (**pztA**, **TKVD**, **zFAF**, **tVDU**)
3. We included failure cases of each domain during human evaluation in Appendix G. (**TKVD**)
4. We provided more domain-specific information on the factual errors that LINK made according to human evaluation. (**pztA**, **TKVD**)
5. We provided more insights on our **distribution approximation method** using stepwise reranker in Appendix C. (**pztA**)
6. We pointed out that reranking the statements for LINK is not the key reason that LINT data can be well separated, but rather reranking the beams **at each step of search** is the key reason, accompanied by ablation studies on post-hoc applying rerankers onto model search results. (**zFAF**)
7. We discussed future work and practical applications of our work. (**pztA**, **zFAF**)
8. We discussed the difference between our work and other works with similar dataset format. (**TKVD**)
9. We discussed the potential sampling bias of our method. (**pztA**)
10. We discussed the scalability of the dataset through meta rules. (**TKVD**)
11. We discussed the principles of creating the meta rules. (**tVDU**)
12. We discussed the reasons for our model choices. (**tVDU**)

We submitted our revised draft and addressed individual concerns. **We hope to have more constructive conversations with the reviewers, so we would appreciate it if the Chairs could help us reach out to the reviewers and follow up on their response.**

**Summary of Contribution**
1. We propose **LINK**, the first framework for systematically generating longtail evaluation data for large language models. We solved the issue of managing longtailness, correctness and scale simultaneously, by grounding the statements on symbolic rules.

- **LINK’s Knowledge Beam Search module** searches for knowledge to populate the symbolic rules by prompting InstructGPT on one variable at a time. Because the generation is grounded on symbolic rules, the statement is guaranteed to be correct as long as we obtain correct values for each variable separately.This process alleviates the pressure to guarantee correctness and longtailness throughout generation.

- **LINK’s critic and reranker modules** verifies the data type correctness and factual correctness of each predicate, and rerankers the values based on log likelihood at each step of the search. They significantly improve the value quality and value distribution of the search result.
2. LINK outperforms prompt-based ChatGPT and GPT4 on **longtail statement generation**.

- We demonstrate this by conducting extensive human evaluation on the generated statements of LINK, ChatGPT and GPT4, as well as detailed analysis on the likelihood distribution of the generated statements.

- We show that using two much smaller language models (Flan-T5-11B and Llama-7B) one can significantly improve the value quality and value distribution of a weaker model (InstructGPT) over stronger models (ChatGPT and GPT4).

- We ablate on the effect of knowledge model, critic and reranker modules by removing each component from LINK and substituting knowledge model with GPT4.

3. We create **LINT** using LINK, a dataset containing 20K+ longtail knowledge statements and 20K+ head knowledge statements. LINT is a valuable resource for researchers in creating more challenging evaluation data.

- LINT reveals a discrepancy of behavior of LLMs on statement distributions. We demonstrate that by analyzing its performance on an entailment classification task on positive and negative statements.

Thank you for your consideration.

Paper 6959 Authors